# Efferocytosis of SARS-CoV-2-infected dying cells impairs macrophage anti-inflammatory functions and clearance of apoptotic cells

Ana CG Salina[1†], Douglas dos-Santos[1†], Tamara S Rodrigues[1†], Marlon Fortes-Rocha[1], Edismauro G Freitas-Filho[1], Daniel L Alzamora-Terrel[1], Icaro MS Castro[2], Thais FC Fraga da Silva[3], Mikhael HF de Lima[4], Daniele C Nascimento[4,5], Camila M Silva[4,5], Juliana E Toller-Kawahisa[4,5], Amanda Becerra[1], Samuel Oliveira[1], Diego B Caetité[4,5], Leticia Almeida[1,5], Adriene Y Ishimoto[1], Thais M Lima[1], Ronaldo B Martins[1], Flavio Veras[5], Natália B do Amaral[6], Marcela C Giannini[6], Letícia P Bonjorno[6], Maria IF Lopes[6], Maira N Benatti[7], Sabrina S Batah[7], Rodrigo C Santana[6], Fernando C Vilar[6], Maria A Martins[8], Rodrigo L Assad[6], Sergio CL de Almeida[6], Fabiola R de Oliveira[6], Eurico Arruda Neto[1], Thiago M Cunha[4,5], José C Alves-Filho[4,5], Vania LD Bonato[3], Fernando Q Cunha[4,5], Alexandre T Fabro[7], Helder I Nakaya[2,5], Dario S Zamboni[1,5], Paulo Louzada-Junior[5,6], Rene DR Oliveira[6], Larissa D Cunha[1*]

[1]Departamento de Biologia Celular e Molecular e Bioagentes Patogênicos, Faculdade de Medicina de Ribeirão Preto, Universidade de São Paulo, Ribeirão Preto, Brazil; [2]Hospital Israelita Albert Einstein, Sao Paulo, Brazil; [3]Departamento de Bioquímica e Imunologia, Faculdade de Medicina de Ribeirão Preto, Universidade de São Paulo, Ribeirão Preto, Ribeirão Preto, Brazil; [4]Departamento de Farmacologia, Faculdade de Medicina de Ribeirão Preto, Universidade de São Paulo, Ribeirão Preto, Brazil; [5]Center of Research in Inflammatory Diseases (CRID), Faculdade de Medicina de Ribeirão Preto, Universidade de São Paulo, Ribeirão Preto, Brazil; [6]Divisão de Imunologia Clinica, Emergência, Doenças Infecciosas e Unidade de Terapia Intensiva, Faculdade de Medicina de Ribeirão Preto, Universidade de São Paulo, Ribeirão Preto, Brazil; [7]Departamento de Patologia e Medicina Legal, Faculdade de Medicina de Ribeirão Preto, Universidade de São Paulo, Ribeirao Preto, Brazil; [8]Departamento de Cirurgia e Anatomia, Faculdade de Medicina de Ribeirão Preto, Universidade de São Paulo, Ribeirão Preto, Brazil

*For correspondence:
larissacunha@usp.br

†These authors contributed equally to this work

Competing interest: The authors declare that no competing interests exist.

**Abstract** COVID-19 is a disease of dysfunctional immune responses, but the mechanisms triggering immunopathogenesis are not established. The functional plasticity of macrophages allows this cell type to promote pathogen elimination and inflammation or suppress inflammation and promote tissue remodeling and injury repair. During an infection, the clearance of dead and dying cells, a process named efferocytosis, can modulate the interplay between these contrasting functions. Here, we show that engulfment of SARS-CoV-2-infected apoptotic cells exacerbates inflammatory cytokine production, inhibits the expression of efferocytic receptors, and impairs continual efferocytosis by macrophages. We also provide evidence supporting that lung monocytes and macrophages from severe COVID-19 patients have compromised efferocytic capacity. Our findings reveal that dysfunctional efferocytosis of SARS-CoV-2-infected cell corpses suppresses macrophage

anti-inflammation and efficient tissue repair programs and provides mechanistic insights for the excessive production of pro-inflammatory cytokines and accumulation of tissue damage associated with COVID-19 immunopathogenesis.

## Editor's evaluation

The lung and other organ damage sustained during Covid-19 disease results predominantly from the immune response to SARS-CoV-2, which includes inflammation. In this paper the authors investigated potential mechanisms, and determined that part of the reason is the interaction of SARS-CoV-2 with macrophages, the cells of the innate immune response which have multiple roles, including efferocytosis, which is the engulfment of apoptotic cells, removing these cells and their debris. They found that SARS-CoV-2 virus is still viable in the apoptotic cells, and leads to an increase in macrophage production of inflammatory cytokines, as well as decreases their efferocytosis capability. In combination, these effects may lead to a more dysregulated immune response to the virus.

## Introduction

Patients with severe and critical COVID-19 progress from pneumonia to development of acute respiratory distress syndrome and respiratory failure, septic shock, and multiorgan dysfunction (*Siddiqi and Mehra, 2020*). These severe clinical manifestations of the disease are caused by dysregulated host immune response, including compromised function of the myeloid compartment (*Blanco-Melo et al., 2020*; *Lucas et al., 2020*; *Rodrigues et al., 2021*; *Schulte-Schrepping et al., 2020*; *Silvin et al., 2020*; *Valle et al., 2020*; *Veras et al., 2020*). Therefore, hyperinflammation and extensive unresolved tissue damage associated with dysfunctional innate responses should contribute to the pathogenesis of inflammatory lung disease and worsening of COVID-19, although underlying mechanisms are still loosely defined.

Macrophages and monocytes are functionally plastic cells that can promote homeostasis and resolution of inflammation by sensing microbial and host-derived signals and programming their gene expression toward an anti-inflammatory, pro-resolution, and wound-healing phenotype (*Martins et al., 2019*; *Trzebanski and Jung, 2020*). This host-protective programming reduces inflammatory cytokine production by other immune and non-immune cells and promotes the repair of damaged tissue to sustain physiological function and re-establish homeostasis (*Murray and Wynn, 2011*). During the repair process, the efficient clearance of dying cells prevents further tissue dysfunction caused by uncontrolled cytotoxicity and the release of damage-associated molecular patterns (DAMP). Defects in sensing, ingesting, and degradation of dead and dying cells through efferocytosis cause chronic inflammation and autoimmune diseases (*Boada-Romero et al., 2020*). In addition to preventing the deleterious effects of secondary necrosis, efferocytosis also couples corpse internalization to other environmental cues (such as local cytokines and metabolites) to temporally and spatially regulate macrophage anti-inflammatory and tissue repair functions (*A-Gonzalez et al., 2017*; *Bosurgi et al., 2017*; *Perry et al., 2019*). While the clearance of apoptotic cells (AC) is often associated with alternative macrophage programming, the identity of the dying cell, the type of cell death, and the context of death (either sterile or infectious) can modulate the nature of the macrophage response (*Rothlin et al., 2021*).

While infection with SARS-CoV-2 induces the recruitment of immune cells to the lungs, their role in host defense and the causes for the dysfunction during disease progression remain elusive. Here, we sought to determine how macrophages operate when responding to dying epithelial cells infected with SARS-CoV-2. We found that the presence of viable SARS-CoV-2 in cell corpses dysregulates macrophage anti-inflammatory responses to the efferocytosis of AC, promoting excessive production of inflammatory IL-6 and IL-1β while disrupting the efficient continual clearance of dead cells required for effective tissue repair. We also provide evidence that the expression of efferocytic genes is reduced in macrophages from severe COVID-19 patients. Therefore, SARS-CoV-2 infection and the clearance of infected dying cells disrupt macrophage host-protective functions associated with immunopathological manifestations of COVID-19.

# Results

## Macrophages engulf dying cells carrying viable SARS-CoV-2

SARS-CoV-2 infection causes cytopathic effects in human and primate epithelial cells (*Chu et al., 2020*; *Zhu et al., 2020*), possibly mediated by activation of multiple cell death pathways (*Chan et al., 2020*; *Li et al., 2020a*; *Li et al., 2020b*; *Mulay et al., 2021*; *Ren et al., 2020*; *Zhu et al., 2020*). Notably, inhibition of apoptosis ameliorates cytokine expression and tissue damage in the lungs of SARS-CoV-2-infected mice, indicating that apoptosis in the lungs is pathogenic (*Chu et al., 2021*). Agreeing with previous reports (*Chu et al., 2020*; *Ren et al., 2020*; *Zhu et al., 2020*), infection with SARS-CoV-2 for 48 hr induced caspase-3 and caspase-8 activation in immortalized simian kidney epithelial Vero CCL81 (*Figure 1A*) and human lung epithelial Calu-3 cell lines (*Figure 1A and B*). We also found features consistent with the predominance of AC death at 48 hr post-infection by flow cytometric analysis of the pattern of annexin V and a permeability dye co-staining in infected cells (*Figure 1C*). Consistently, we observed low levels of cytolysis in these infected cells, as assessed by lactate dehydrogenase release (*Figure 1D*). Immunofluorescent labeling of cleaved caspase-3 in lung tissues of deceased COVID-19 patients confirmed the induction of apoptosis in the epithelia infected with replicant SARS-CoV-2 (*Figure 1E*, and *Figure 1—figure supplement 1A*). Examination of lung tissues also revealed evidence of macrophages with internalized epithelial cells and SARS-CoV-2, suggesting the uptake of infected epithelial cells by macrophages (*Figure 1F*).

We performed the median tissue culture infection dose (TCID50) assay and found that viral particles obtained from isolated apoptotic Vero and Calu-3 cells induced cytopathic effect to an equivalent extent as those released in the culture supernatants (*Figure 1—figure supplement 1B*). Furthermore, we isolated annexin V-labeled AC from SARS-CoV-2-infected cell culture and confirmed that they carry viable viral particles (*Figure 1G*). Therefore, viable SARS-CoV-2 is retained within infected, apoptotic epithelial cells.

Exposure of phosphatidylserine (PtdSer) on the outer plasma membrane of cells undergoing regulated cell death, as observed by annexin V binding to infected dying cells (*Figure 1C*), is the most ubiquitous signal that triggers their phagocytosis (*Fadok et al., 1992*; *Nagata, 2018*). To determine whether phagocytes efficiently engulf SARS-CoV-2-infected AC (CoV2-AC), we collected the loosely attached AC from infected cultures and used flow cytometry to assess their uptake by primary human macrophages and macrophages differentiated from THP-1 monocytes. We observed that macrophages efficiently engulf CoV2-AC, similarly to the uptake of sterile, UV-irradiated AC (UV-AC) (*Figure 1H* and *Figure 1—figure supplement 1C*). Analysis by confocal microscopy and staining for Spike protein confirmed the presence of SARS-CoV-2 in engulfed cell corpses (*Figure 1—figure supplement 1D*). Finally, either stimulation with infected or sterile dying cells did not robustly affect macrophages viability up to 24 hr post-treatment (*Figure 1—figure supplement 1E*).

These results demonstrate that macrophages engulf cell corpses carrying viable particles of SARS-CoV-2, offering a framework to investigate their effects on macrophage function.

## Efferocytosis of SARS-CoV-2-infected dying cells impairs macrophage anti-inflammatory function

We next addressed the effect of the engulfment of SARS-CoV-2-infected AC in macrophages. The uptake of sterile AC often promotes macrophage functional polarization toward an anti-inflammatory and tissue repair phenotype (*Doran et al., 2020*). We first observed that stimulation of primary monocyte-derived macrophages with CoV2-AC, but not infection with SARS-CoV-2, reduced the expression of genes associated with alternative programming to tissue remodeling and secretion of immune-modulatory mediators such as *CCL18*, *CD206* (also known as *MRC1*), *MMP9*, *PPARG*, and *CD163* (*Figure 2A*). Furthermore, stimulation with UV-AC obtained from infected Vero cells increased both gene (*Figure 2A*) and surface protein (*Figure 2B*, *Figure 2—figure supplement 1A*) expression of the anti-inflammatory marker CD206. However, upregulation of *MRC1* at the gene or protein levels did not occur in response to stimulation with CoV2-AC (*Figure 2A and B*). We obtained similar results in THP-1-derived macrophages stimulated with CoV2-AC obtained from either Calu-3 or Vero cells infected with SARS-CoV-2 (*Figure 2—figure supplement Figure 2—figure supplement 1B-D*). Stimulation with the conditioned supernatants of infected dying cells (containing putative DAMP, cytokines produced by epithelial cells, extracellular vesicles, and released virions) or direct infection with SARS-CoV-2 did not induce *MRC1* transcription (*Figure 2A*, *Figure 2—figure supplement 1B*) or

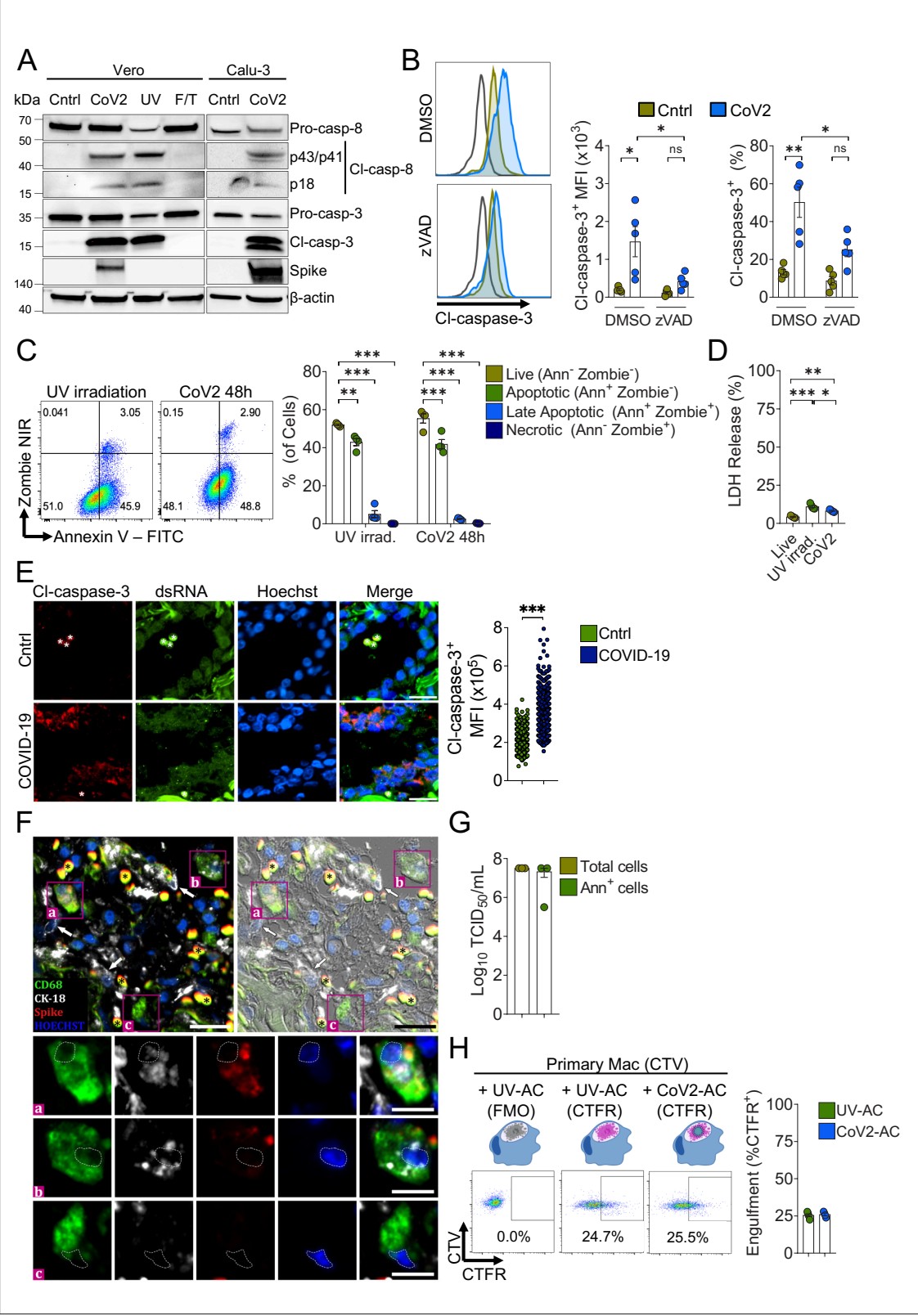

**Figure 1.** Macrophages engulf apoptotic cells (AC) carrying viable SARS-CoV-2. (**A**) Immunoblot of caspase-8 and caspase-3 cleavage in Calu-3 and Vero CCL81 cells in response to infection with SARS-CoV-2 for 48 hr (CoV2, detected with rabbit anti-Spike), UV irradiation (UV), or freezing and thawing (F/T). (**B**) Flow cytometric analysis of caspase-3 cleavage (Cl-caspase-3) in Calu-3 cells unstimulated (Cntrl) or infected with CoV2 for 48 hr. The pan-caspase inhibitor z-VAD-FMK (zVAD; 20 μM) or vehicle (DMSO) was added to cell cultures after viral adsorption. Representative histogram, geometric

*Figure 1 continued on next page*

*Figure 1 continued*

mean fluorescence intensity (MFI), and frequency of Cl-caspase-3$^+$ cells (gated on total cells/single cells) are shown. The black line on the histogram represents the control sample labeled with secondary antibody only. (**C**) Flow cytometric analysis of apoptosis in Vero CCL81 cells in response to UV light irradiation or 48 hr post-infection with CoV2, assessed by annexin V (FITC) and viability dye (Zombie NIR) co-staining. Representative histogram and frequency of each population (pre-gated on total cells/single cells) are shown. (**D**) Cytotoxicity analysis by measurement of lactate dehydrogenase (LDH) release on the supernatants of Vero CCL81 cells in response to UV light irradiation or 48 hr post-infection with CoV2. (**E, F**) Representative histological findings in post-mortem lung tissue from COVID-19 patients obtained by ultrasound-guided minimally invasive autopsy. (**E**) Immunofluorescence images of tissue samples labeled with anti-Cl-caspase-3 (red), anti-dsRNA (green), and stained with Hoechst (nuclei, blue) for the detection of caspase-3 activation in infected epithelia (COVID-19, bottom panels) compared to samples from control patients (Cntrl, upper panels). Tissues were scanned by wide-field epifluorescence imaging. Scale bar: 20 μm, asterisks: erythrocytes. The MFI of Cl-caspase-3 of at least 300 cells is shown (n=3 individuals per group). Technical control for secondary antibodies background is shown in *Figure 1—figure supplement 1*. (**F**) Representative immunofluorescence image of a COVID-19 patient tissue sample labeled for the detection of efferocytosis in situ and superimposed with differential interference contrast. Samples were labeled with anti-CD68 (detecting macrophages, green), anti-cytokeratin18 (CK-18, detecting epithelial cells, white), anti-Spike (detecting SARS-CoV-2, red), and stained with Hoechst (nuclei, blue). Bottom panels show higher magnification of ( a,b) CD68$^+$ macrophage with CK-18 and Spike labeling in the cytosol and (c) CD68$^+$ only macrophage. White arrows: CK-18$^+$ cells; asterisks: erythrocytes. Tissues were scanned by wide-field epifluorescence imaging. Scale bar: 20 μm; scale bar for insets: 10 μm. (**G**) Quantification by TCID$_{50}$ of SARS-CoV-2 viral loads in Vero CCL81 cells infected for 48 hr. Viral loads were estimated for pre-sorted (total cells) or annexin V$^+$ (Ann$^+$ cells) cells isolated by magnetic separation. (**H**) Uptake of apoptotic Vero CCL81 cells (labeled with CTFR) in response to UV irradiation (UV-AC) or infection with SARS-CoV-2 for 48 hr (Cov2-AC) by human monocyte-derived macrophages (labeled with CellTrace Violet (CTV)). Macrophages were co-incubated with AC for 2 hr, and internalization was assessed by flow cytometry. Representative plot and percentage of engulfment (gated on total cells/single cells/live cells/CTV$^+$/ CTV$^+$CTFR$^+$ cells) are shown. Boxes represent the mean of five (**B, D**), four (**C, H**), or three (**G**) biological replicates, and error bars are ± S.E.M. Each biological replicate is shown as a circle. Significance was calculated by ANOVA (**B–D**), Mann-Whitney test (**E**), or Student's t-test (**G, H**). *p<0.05, **p<0.001, ***p<0.0001, comparing the indicated groups. Data shown are from one representative out of two (**A, C, G**) or three (**B, D, H**) experiments performed independently with similar results.

The online version of this article includes the following source data and figure supplement(s) for figure 1:

**Source data 1.** Source data for *Figure 1*.

**Figure supplement 1.** Support data for *Figure 1*.

**Figure supplement 1—source data 1.** Source data for *Figure 1—figure supplement 1*.

surface expression (*Figure 2B*, *Figure 2—figure supplement Figure 2—figure supplement 1C and D*) in macrophages. We also confirmed that modulation of CD206 expression occurred specifically in the macrophages that engulfed AC and was not due to paracrine signaling (*Figure 2C* and *Figure 2—figure supplement 1E*). To determine if viral viability was required to modulate CD206 expression by infected AC, isolated CoV2-AC were exposed to UV irradiation for virus inactivation (*Figure 2—figure supplement 1F*). Notably, macrophages treated with epithelial AC containing inactivated SARS-CoV-2 exhibited increased CD206 surface expression, similar to that of UV-AC (*Figure 2D*). This finding suggests that viable SARS-CoV-2 actively represses alternative programming carried in dying cells and is not primarily enforced by the type of cell death caused by the infection. Efferocytosis of sterile UV-AC in the presence of SARS-CoV-2 still induced higher expression of CD206 in macrophages (*Figure 2E*). Therefore, the suppression of CD206 surface expression required delivery of viral particles within cell corpses to macrophages.

Conversely, the uptake of CoV2-AC, but not UV-AC, significantly increased *IL6* expression in monocyte- and THP-1-derived macrophages (*Figure 3A* and *Figure 3—figure supplement 1A*, respectively). The uptake of CoV2-AC also induced robust secretion of inflammatory IL-6 and IL-1β, in both primary (*Figure 3B*) and THP-1-derived macrophages (*Figure 3—figure supplement 1B and C*). We did not detect these cytokines in macrophages stimulated with the cell-free conditioned supernatants (*Figure 3B* and *Figure 3—figure supplement 1A-C*), and therefore this effect was not due to immune mediators or viral particles released from infected epithelial cells. Importantly, assessment of IL-6 secretion in macrophages stimulated with sorted out populations of infected dying cells revealed that cytokine production occurred specifically in response to AC, but not permeabilized dead cells (*Figure 3C*). In agreement, CoV2-AC obtained from epithelial cells infected in the presence of osmo-protectant glycine, which non-specifically prevents the leak of cytosolic content by lytic permeabilization (*Frank et al., 2000*; *Evavold et al., 2018*) still induced robust secretion of IL-6 by macrophages (*Figure 3—figure supplement 1D*).

Further, the induction of inflammatory cytokines by SARS-CoV-2-loaded corpses required the presence of viable viral particles, as UV treatment or paraformaldehyde (PFA) fixation of isolated

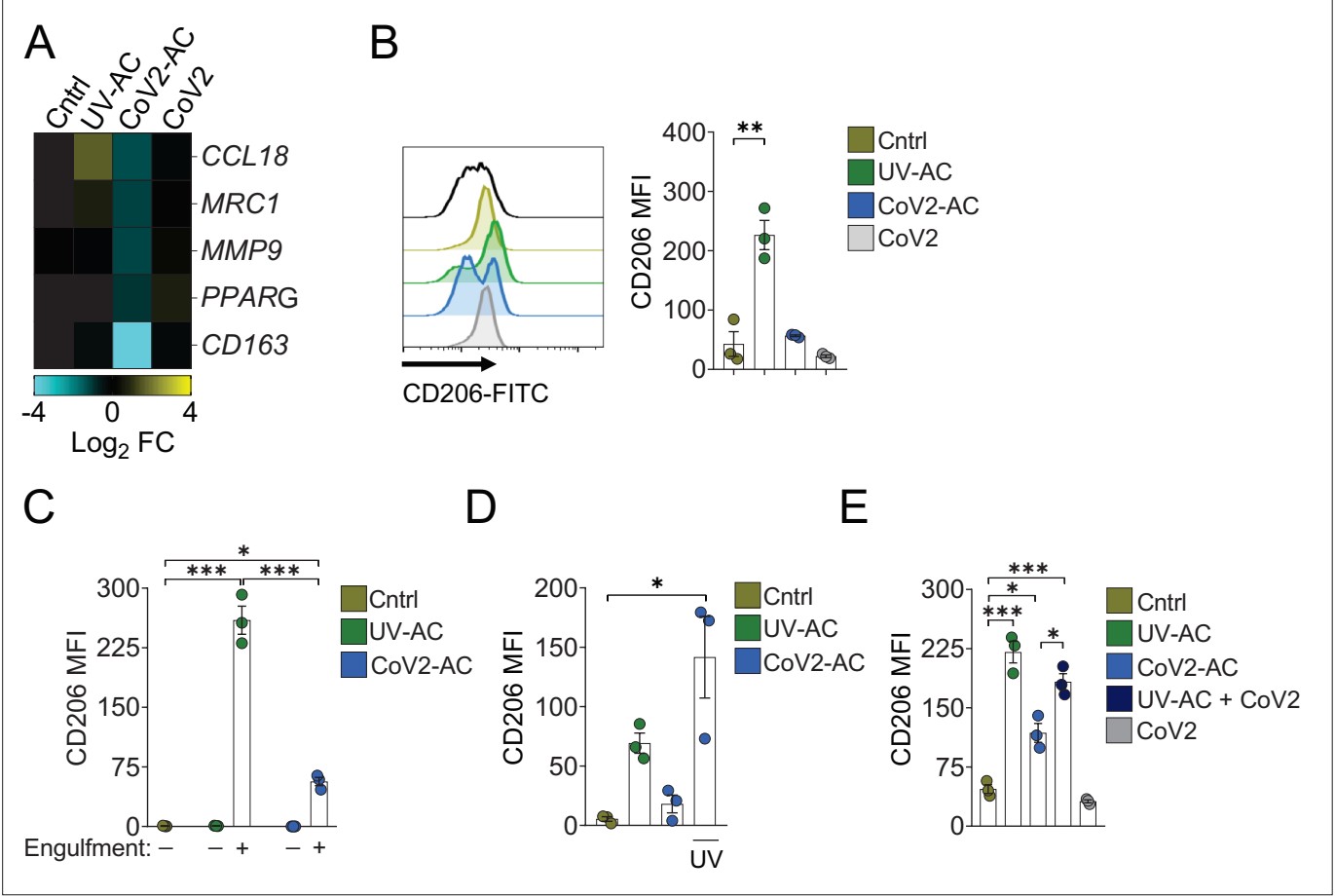

**Figure 2.** Engulfment of SARS-CoV-2-infected dying cells impairs macrophage anti-inflammatory functions. Macrophages were unstimulated (Cntrl), co-incubated with apoptotic Vero CCL81 cells (AC) isolated from UV-irradiated (UV-AC) or SARS-CoV-2-infected (CoV2-AC) cultures, stimulated with the supernatants of infected apoptotic cells (CoV2-AC Sup), or infected with CoV2 (CoV2), as indicated. (**A**) Heatmap showing the expression of M2-marker genes (*CCL18*, *MRC1*, *MMP9*, *PPARG,* and *CD163*) in human monocyte-derived macrophages 24 hr after stimulation, calculated as log$_2$ fold change relative to Cntrl. mRNA expression was determined by RT-qPCR and normalized to *GAPDH*. Data represent the mean of biological triplicates. (**B**) Flow cytometric analysis of CD206 expression on the cell surface of human monocyte-derived macrophages (gated on total cells/single cells/live cells; gating strategy in *Figure 2—figure supplement 1A*) 24 hr after stimulation. Representative histograms and quantification of geometric mean fluorescence intensity (MFI) are shown. The black line on the histogram represents the fluorescence minus one (FMO) control. (**C–E**) Flow cytometric analysis of CD206 expression on the cell surface of THP-1-derived macrophages 24 hr after stimulation, showing quantification of CD206 MFI. (**C**) Macrophages (labeled with CTFR) were stimulated with UV-AC or CoV2-AC (labeled with CTV), and CD206 expression was measured on macrophages that engulfed (+) or not (−) AC. Gating strategy is depicted in *Figure 2—figure supplement 1E*. (**D**) Where indicated, isolated CoV2-AC were UV-irradiated for 20 min prior to co-incubation for viral inactivation. (**E**) The responses of macrophages to UV-AC and CoV2-AC were compared to infection with CoV2 in the presence of UV-AC (UV-AC + CoV2). Boxes represent the mean of three biological replicates using cells from a single donor (**A, B**) or THP-1-derived cells (**C–E**), and error bars are ± S.E.M. Each biological replicate is shown as a circle. Significance was calculated by ANOVA. *p<0.05, **p<0.001, ***p<0.0001, comparing the indicated groups. Data shown are from one representative out of at least two experiments performed independently with similar results.

The online version of this article includes the following source data and figure supplement(s) for figure 2:

**Source data 1.** Source data for *Figure 2*.

**Figure supplement 1.** Support data for *Figure 2*.

**Figure supplement 1—source data 1.** Source data for *Figure 2—figure supplement 1*.

CoV2-AC abrogated IL-6 secretion (*Figure 3D*). Notably, we did not observe robust production of IL-6 in response to dying epithelial cells infected with Coxsackievirus (*Figure 3—figure supplement 1E* and *Figure 3E*). This result suggests that augmented pro-inflammatory cytokine production is not a universal response of macrophages to the uptake of cell corpses infected with positive single-strand viruses.

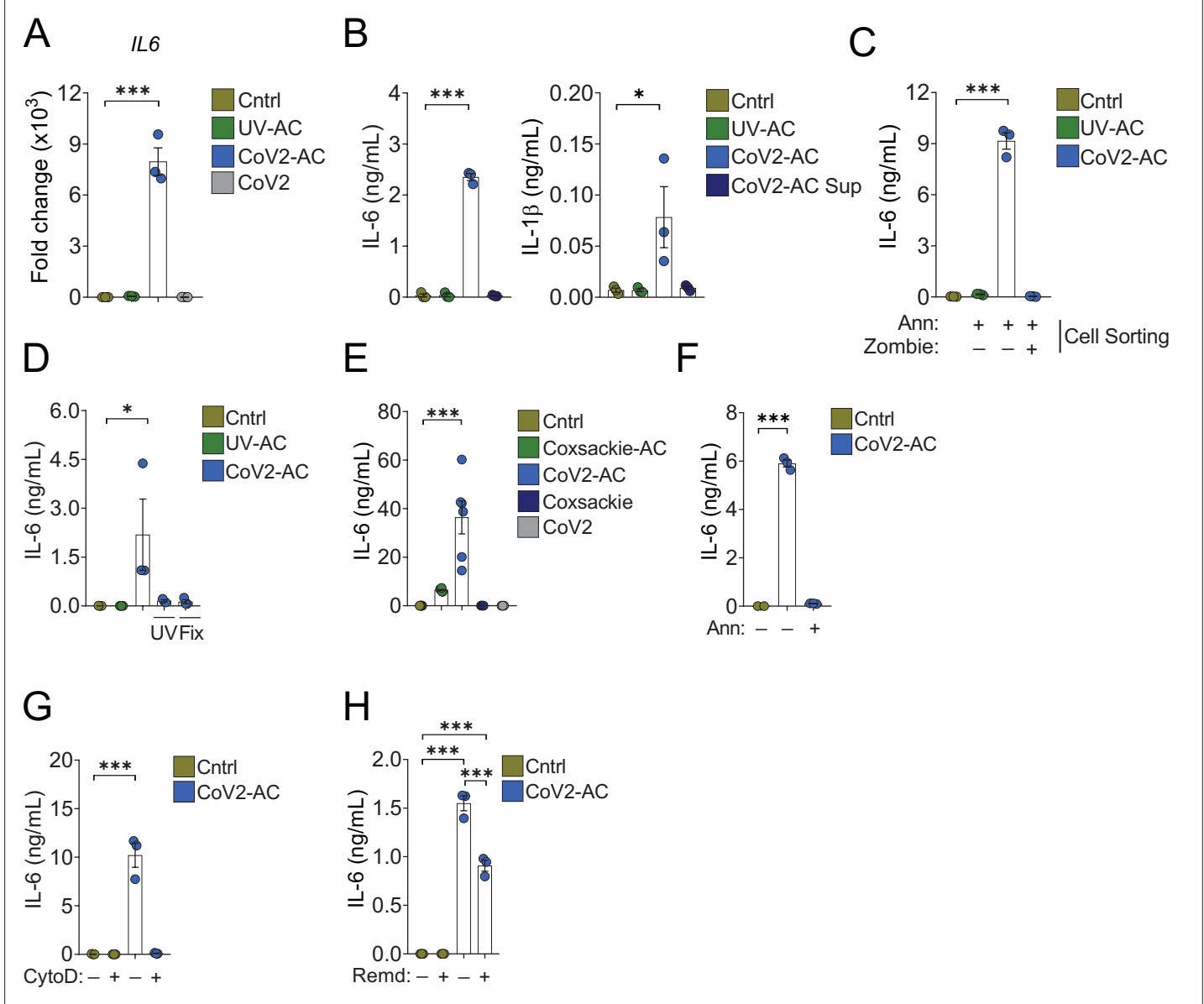

**Figure 3.** Engulfment of SARS-CoV-2-infected dying cells causes inflammatory cytokine production. Macrophages were left unstimulated (Cntrl), co-incubated with apoptotic Vero CCL81 cells (AC) isolated from UV-irradiated (UV-AC) or SARS-CoV-2-infected (CoV2-AC) cultures, stimulated with the supernatants of infected apoptotic cells (CoV2-AC Sup), or infected with CoV2, as indicated. (**A**) *IL6* expression in primary human monocyte-derived macrophages 24 hr after stimulation showed as fold change relative to Cntrl. mRNA expression was determined by RT-qPCR and normalized to *GAPDH*. (**B**) Cytometric bead array quantification of IL-6 and IL-1β in the culture supernatants of the monocyte-derived macrophages 24 hr after stimulation. (**C, D**) ELISA quantification of IL-6 in the culture supernatants of THP-1-derived macrophages 24 hr after stimulation. (**C**) Macrophages were stimulated with early AC (annexin V+ Zombie-) or late AC (annexin V+ Zombie+) sorted out by flow cytometry. (**D**) Where indicated, isolated CoV2-AC were UV-irradiated (UV) for 20 min or fixed with 2% paraformaldehyde (Fix) for 10 min prior to co-incubation. (**E–H**) ELISA quantification of IL-6 in the culture supernatants of human primary monocyte-derived macrophages 24 hr after stimulation. (**E**) IL-6 secretion in response to CoV2-AC was compared to AC isolated from Vero CCL81 cells similarly infected with Coxsackievirus (Coxsackie-AC). (**F**) Phosphatidylserine on the surface of CoV2-AC was blocked by incubation with annexin V (Ann, 0.1 μg/mL) prior to addition to macrophage cultures to inhibit AC binding. (**G**) CoV2-AC and macrophages were co-incubated in the presence of cytochalasin D (Cyto D, 10 μM) to inhibit AC internalization. (**H**) Macrophages were treated with Remdesivir (Remd, 20 μM) following stimulation with CoV2-AC to inhibit viral RNA transcription. Boxes represent the mean of three biological replicates using cells from a single donor (**A, B, F, G**), two donors (**E**), or THP-1-derived cells (**C, D**), and error bars are ± S.E.M. Each biological replicate is shown as a circle. Significance was calculated by ANOVA. *p<0.05, **p<0.001, ***p<0.0001, comparing the indicated groups. Data shown are from one representative out of at least two experiments performed independently with similar results.

The online version of this article includes the following source data and figure supplement(s) for figure 3:

*Figure 3 continued*

**Source data 1.** Source data for *Figure 3*.

**Figure supplement 1.** Support data for *Figure 3*.

**Figure supplement 1—source data 1.** Source data for *Figure 3—figure supplement 1*.

To confirm the requirement of recognition and binding of CoV2-AC to promote inflammatory skewing, we stimulated macrophages in the presence of annexin V to mask PtdSer exposed on the surface of the AC. In the absence of PtdSer ligation, macrophages did not secrete IL-6 in response to CoV2-AC (*Figure 3F* and *Figure 3—figure supplement 1F*). Stimulation of macrophages in the presence of actin depolymerization inducer Cytochalasin D, which allows corpse binding but not internalization, also attenuated IL-6 production (*Figure 3G*). Finally, we tested the effect of antiviral drugs targeting viral RNA transcription (RdRP-mediated RNA synthesis). We found that Remdesivir treatment of primary macrophages following stimulation with CoV2-AC partially reduced IL-6 secretion, suggesting that viral sgRNA expression is important to modulate the inflammatory response (*Figure 3H*).

Collectively, these findings support that sensing and engulfment of dying cells carrying viable SARS-CoV-2 switch macrophage anti-inflammatory, resolutive programming in response to efferocytosis toward an inflammatory phenotype. The exacerbated cytokine production observed in response to the efferocytosis of infected cell corpses by macrophages may contribute to the cytokine storm associated with COVID-19 hyperinflammatory syndrome.

## Efferocytosis of SARS-CoV-2-infected dying cells suppresses continual clearance of apoptotic cells

Several receptors mediate efferocytosis through recognition of PtdSer on the surface of a dying cell, either by direct binding or through a bridging molecule (*Boada-Romero et al., 2020*; *Penberthy and Ravichandran, 2016*). We found that engulfment of CoV2-AC by primary macrophages reduced the transcription of such PtdSer receptors, including the scavenger receptors *CD36* and *SRA1*, αVβ5 integrin (*ITGB5*), and T cell immunoglobulin mucin receptor 4 (*TIM4*) and MER proto-oncogene tyrosine kinase (*MERTK*) (*Figure 4A*). In THP-1-macrophages, we also observed reduced expression of *SRA1*, *ITGB5*, and *TIM4* receptors in response to infection with SARS-CoV-2 (*Figure 4—figure supplement 1A*). Phagocytes can ingest multiple corpses in subsequent rounds of efferocytosis (*Miyanishi et al., 2007*; *Morioka et al., 2018*; *Park et al., 2011*; *Yurdagul et al., 2020*). Previous in vivo work showed that macrophages must continually remove ACs to promote efficient repair of injury and prevent the accumulation of secondarily necrotic cells (*Wang et al., 2017*; *Yurdagul et al., 2020*). To determine if repression of efferocytic receptors affects additional uptake of dying cells, we treated macrophages with CoV2-AC and subsequently fed them with apoptotic human Jurkat cells (UV-Jurkat). We found that the engulfment of SARS-CoV-2-infected corpses suppressed the efferocytosis of other AC (*Figure 4B–E*, *Figure 4—figure supplement 1B*). As infection with SARS-CoV-2 reduced the expression of some efferocytic receptors in THP-1 macrophages (*Figure 4—figure supplement 1A*), we also tested if it affected the uptake of AC. Comparatively, infection with SARS-CoV-2 reduced AC clearance to a lower extent than prior uptake of CoV2-AC (*Figure 4—figure supplement 1C*). Notably, treatment with Remdesivir did not improve the subsequent uptake by macrophages stimulated with CoV2-AC, arguing that viral RNA replication is not required for suppression of efferocytosis in response to CoV2-AC (*Figure 4F*). This result also suggests that different mechanisms lead to pro-inflammatory cytokine production and efferocytosis repression in response to the uptake of infected corpses.

Thus, efferocytosis of SARS-CoV-2-infected dying cells affects the expression of efferocytic receptors and impairs the continual removal of AC by macrophages.

## Lung monocytes and macrophages of severe COVID-19 patients express reduced levels of efferocytic receptors

To gain insight into the contribution of dysfunctional efferocytosis in COVID-19 pathogenesis, we assessed the expression of efferocytic receptors by immunofluorescence in lung tissues obtained from autopsies of deceased COVID-19 patients. We found a reduction in the protein levels of CD36 in S1009[+] infiltrating phagocytes in the lungs of COVID-19 patients compared to control tissues

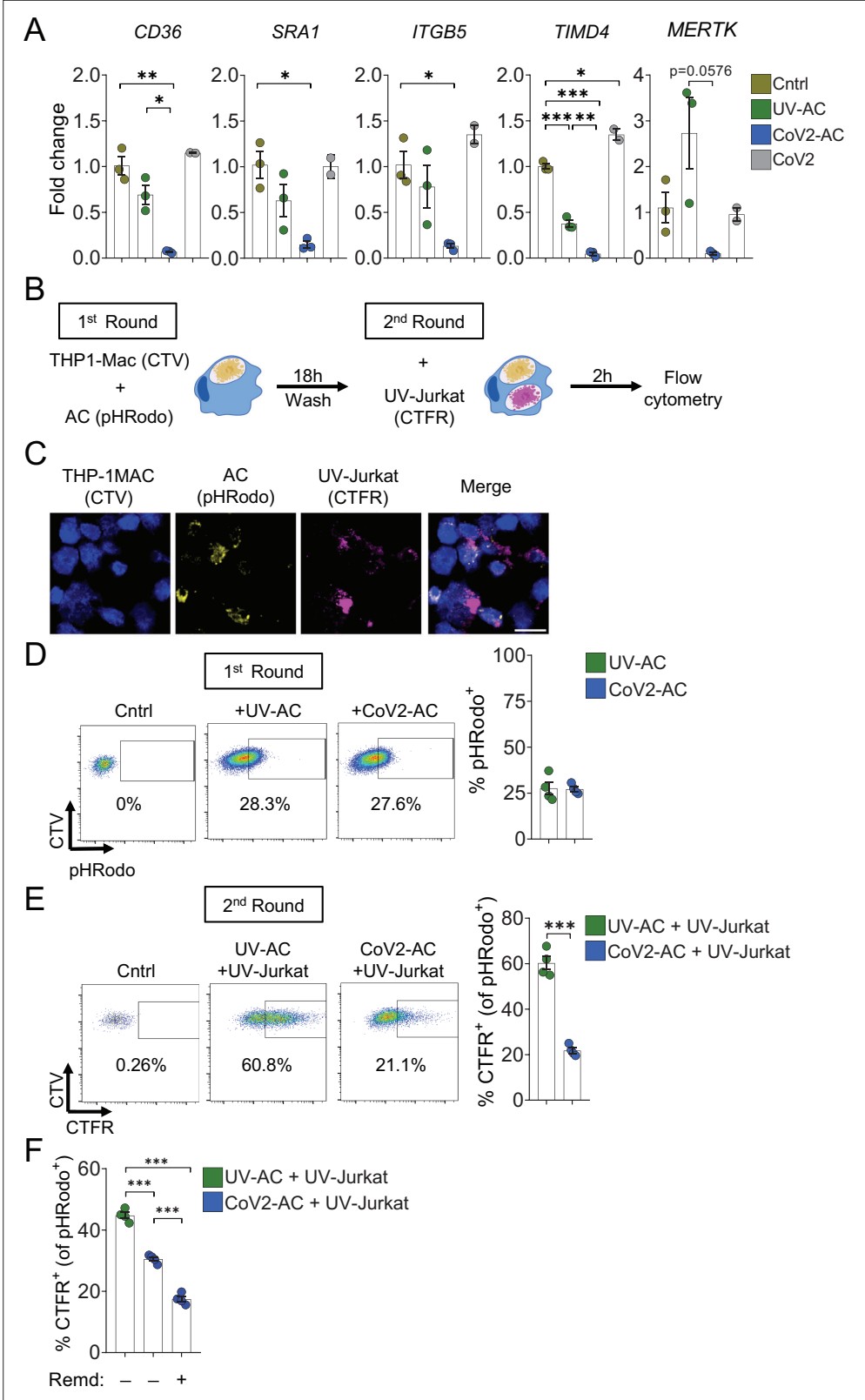

**Figure 4.** Engulfment of SARS-CoV-2-infected dying cells suppresses continual efferocytosis by macrophages.
(**A**) Expression of efferocytic receptors in human primary monocyte-derived macrophages. Macrophages were left
unstimulated (Cntrl), co-incubated for 24 hr with apoptotic Vero CCL81 cells (AC) isolated from UV-irradiated (UV-
AC) or SARS-CoV-2-infected (CoV2-AC) cultures, or infected with CoV2 (CoV2), as indicated. Expression of *CD36*,

*Figure 4 continued on next page*

*Figure 4 continued*

*SRA1*, *ITGB5*, *TIMD4*, and *MERTK* is showed as fold change relative to Cntrl. mRNA levels were determined by RT-qPCR and normalized to *GAPDH*. (**B–F**) Flow cytometric analysis of two-step efferocytosis in vitro. THP-1-derived macrophages (CTV-labeled) were incubated with UV-AC or CoV2-AC isolated from Vero CCL81 cell cultures (labeled with pHRodo) for 18 hr, and subsequently incubated with UV-irradiated apoptotic Jurkat cells (UV-Jurkat, CTFR-labeled) for 2 hr. (**B**) Schematic representation of consecutive co-incubations of THP-1-derived macrophages with AC prior to the assessment of cell corpse uptake by flow cytometry. (**C**) Representative maximal projection of scanning confocal images showing the uptake of AC (pHRodo) and UV-Jurkat (CTFR) by macrophages (CTV) after consecutive co-incubations. (**D**) Percentage of macrophages with internalized UV-AC or CoV2-AC (gated on total cells/single cells/live cells/CTV$^+$/CTV$^+$ pHRodo$^+$); gating strategy is displayed in *Figure 4—figure supplement 1B*. (**E**) Percentage of macrophages that engulfed UV-Jurkat within pHRodo$^+$ populations (gated on total cells/single cells/live cells/CTV$^+$/pHRodo$^+$/pHRodo$^+$ CTFR$^+$). (**F**) Percentage of macrophages that engulfed UV-Jurkat within pHRodo$^+$ populations (gates as in (**E**)) wherein macrophages were treated with Remdesivir (Remd, 20 µM) following stimulation with CoV2-AC. Boxes represent the mean of three (**A**) or four (**D–F**) biological replicates using cells from a single donor in (**A**) or THP-1-derived macrophages (**D–F**), and error bars are ± S.E.M. Each biological replicate is shown as a circle. Significance was calculated by ANOVA (**A, F**) or Student's t-test (**D, E**). *p<0.05, **p<0.001, ***p<0.0001, comparing the indicated groups. Data shown are from one representative out of three experiments performed independently with similar results.

The online version of this article includes the following source data and figure supplement(s) for figure 4:

**Source data 1.** Source data for *Figure 4*.

**Figure supplement 1.** Support data for *Figure 4*.

**Figure supplement 1—source data 1.** Source data for *Figure 4—figure supplement 1*.

---

(*Figure 5A*). Phagocytes in the lungs of COVID-19 patients also expressed lower protein levels of MERTK (*Figure 5B*).

Using publicly available single-cell RNA sequencing (scRNA-seq) data from bronchoalveolar lavage (BAL) (*Liao et al., 2020*), we performed enrichment analysis using the genes differentially expressed in mild and severe COVID-19 patients compared with healthy individuals. We targeted efferocytosis-related gene ontology (GO) annotated pathways and customized gene sets based on the literature (*Boada-Romero et al., 2020*; *Penberthy and Ravichandran, 2016*; *Supplementary file 1*). Independent analysis of clusters previously assigned as macrophages (*Liao et al., 2020*) revealed significant repression of gene sets related to efferocytosis in severe patients (*Figure 5C*). While differences were evident between moderate and severe patients in clusters expressing markers of early infiltrating phagocytes (S1009$^+$ CCL18$^-$), they were more pronounced in mature, likely anti-inflammatory macrophages (S11009$^-$ CCL18$^+$). Notably, reassessment on scRNAseq by *Ren et al., 2021* showed enrichment in pathways related to efferocytosis for genes repressed in alveolar macrophages which were positive for SARS-CoV-2 RNA compared with those which were negative for the virus (*Figure 5D*). This finding is consistent with the notion that uptake of infected cells reduces the capacity of macrophages to clear other dead cells. Altogether, these data support that macrophages in the lungs of severe COVID-19 patients may also fail in their efferocytic capacity.

## Discussion

Patients with severe COVID-19 develop life-threatening inflammatory process in the lungs with underlying causes still not established. Here, we demonstrate that the sensing and engulfment of SARS-CoV-2-infected dying cells by macrophages switch the effector response to efferocytosis from a potential wound healing, anti-inflammatory function to a pro-inflammatory one. Our data supports that the uptake of cell corpses infected with SARS-CoV-2 exacerbates the secretion of inflammatory IL-6 and IL-1β, suggesting a mechanism for the robust secretion of cytokines related to COVID-19 cytokine storm. Another important consequence of this shift in macrophage function is the impairment of macrophage capacity to engulf AC continually and promote proper injury resolution. Instead, our data suggest that efferocytosis of infected cells may augment tissue damage by causing inefficient clearance of dead cells. This process may consequently contribute to respiratory complications developed by patients with the severe form of the disease and increase susceptibility to secondary bacterial infections for the lack of effective disease tolerance mechanisms that restrain collateral tissue damage (*Jamieson et al., 2013*). While such causal connection remains to be investigated, our histological

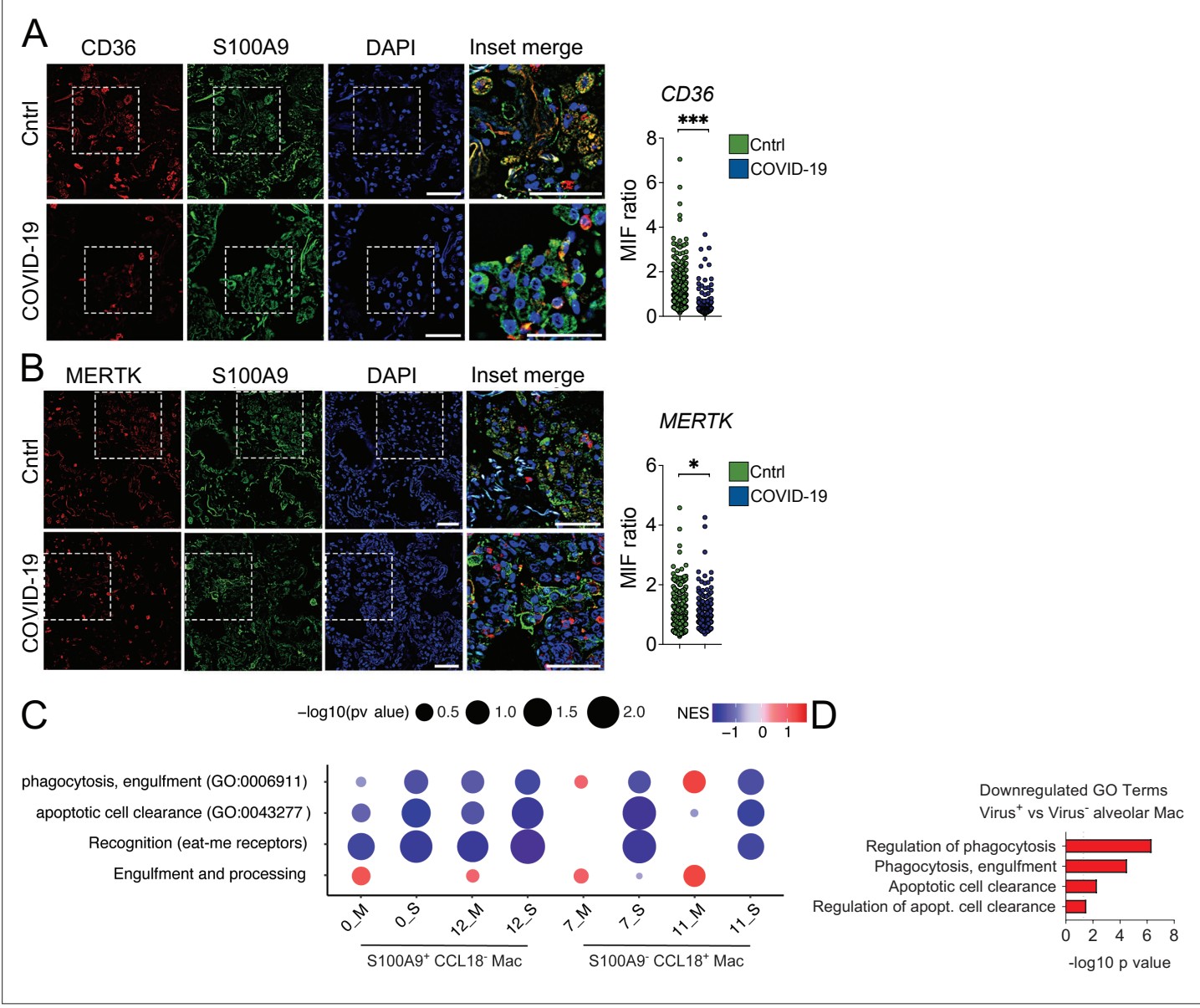

**Figure 5.** Lung monocytes and macrophages of severe COVID-19 patients express reduced levels of efferocytic receptors. (**A–B**) Representative histological findings in post-mortem lung tissue from control patients (Cntrl) and COVID-19 patients, obtained by ultrasound-guided minimally invasive autopsy. Tissue samples were immunolabeled with anti-S100A9 (phagocytes, green), (**A**) anti-CD36 (red), or (**B**) anti-MERTK (red) and stained with DAPI (nuclei, blue). Representative images show cropped details of lung tissues scanned by wide-field epifluorescence imaging. Scale bar: 10 µm. The mean fluorescence intensity (MFI) of CD36 and MERTK of at least 300 S1009+ cells are shown (n=6 control and four COVID-19 patient samples). Dots represent an S1009+ cell (**A and B**), the crossing line represents the mean, and error bars are ± S.E.M. Significance was calculated by Mann-Whitney test (**A and B**), *p<0.05, **p<0.001, ***p<0.0001, comparing the indicated groups. (**C**) Gene set enrichment analysis for efferocytosis-related gene sets in early infiltrating phagocytes (S1009+ CCL18-) and in anti-inflammatory monocytes-derived macrophages (CD14+ S11009- CCL18+) clusters from the bronchoalveolar lavage of mild (-M Figure) and severe (-S) COVID-19 patients versus healthy individuals. The corrplot depicts the normalized enrichment score (NES), and p value for the gene sets indicated on the y-axis. (**D**) Enriched efferocytosis-related gene ontology (GO) terms in genes repressed in virus-positive versus virus-negative alveolar macrophages.

The online version of this article includes the following source data for figure 5:

**Source data 1.** Source data for *Figure 5*.

assessment and reanalysis of scRNAseq datasets suggest that the efferocytic capacity of lung macrophages is impaired in the lungs of COVID-19 patients.

In the context of COVID-19 immunopathogenesis, excessive amounts of necrotic cells may lead to high levels of circulating DAMP, such as HMGB1 and lactate dehydrogenase (LDH), both of which correlate with disease severity (*Chen et al., 2020*; *Han et al., 2020*; *Rodrigues et al., 2021*). It is thus possible that the combination of exacerbated cytokine production and interruption of continual efferocytosis in response to engulfment of SARS-CoV-2-infected corpses could both increase the magnitude and duration of inflammation, contributing to the hyperinflammatory state and multiorgan damage in COVID-19. Finally, some studies reported higher levels of autoantibodies (such as against IFNγ, phospholipids, and annexin A2) in COVID-19 patients (*Bastard et al., 2020*; *Zuniga et al., 2021*; *Zuo et al., 2020*), suggesting that autoantibodies may drive the worsening of the disease. It has long been known that defective engulfment or processing of dying cells causes the development of autoimmune diseases (*Boada-Romero et al., 2020*; *Cohen et al., 2002*; *Miyanishi et al., 2007*; *Peng and Elkon, 2011*; *Rodriguez-Manzanet et al., 2010*). In light of our findings, perhaps defective clearance of AC caused by SARS-CoV-2 infection and SARS-CoV-2-loaded ACs also contributes to COVID-19-associated autoimmunity.

As it becomes clear that worsening of COVID-19 occurs because of immune dysfunction, antivirals, and other therapeutic efforts to limit viral replication may fail to benefit critically ill patients. In those patients, successful therapeutic strategies may rely on targeting dysregulated components of the host response to limit damage to the host, thus promoting disease tolerance and stimulating a resolutive response that restores homeostasis. In this context, our study provides important insights into the mechanisms driving the pathophysiology of COVID-19 that can be explored in the design of therapeutic approaches toward harnessing innate immune responses and macrophage function during the evolution of SARS-CoV-2 infection.

## Materials and methods

**Key resources table**

| Reagent type (species) or resource | Designation | Source or reference | Identifiers | Additional information |
|---|---|---|---|---|
| Antibody | anti-Sars-CoV-2 Spike (rabbit polyclonal) | Abcam | Cat: #ab272504 RRID: AB_2847845 | WB and FACS (1:1000); Immunofluorescence (1:1000) |
| Antibody | anti-CD11b+CD11 c (mouse monoclonal) | Abcam | Cat: #ab1211 RRID: AB_442947 | Immunofluorescence (1:300) |
| Antibody | anti-human CD68 (mouse monoclonal) | Abcam | Cat: #ab955 RRID: AB_307338 | Immunofluorescence (1:100) |
| Antibody | anti-CD206-FITC (mouse monoclonal) | BD Biosciences | Cat: #551135 RRID: AB_394065 | FACS (1:200) |
| Antibody | anti-caspase-3 (rabbit polyclonal) | Cell Signaling | Cat: #9662 RRID: AB_331439 | WB (1:1000) |
| Antibody | anti-cleaved caspase-3 (rabbit monoclonal) | Cell Signaling | Cat: #9664 RRID: AB_10694088 | WB and Immunofluorescence (1:1000); FACS (1:200) |
| Antibody | anti-caspase-8 (rabbit monoclonal) | Cell Signaling | Cat: #4790 RRID: AB_10545768 | WB (1:1000) |
| Antibody | anti-cleaved caspase-8 (rabbit monoclonal) | Cell Signaling | Cat: #9496 RRID: AB_561381 | WB (1:1000) |
| Antibody | anti-βactin HRP conjugate (mouse monoclonal) | Cell Signaling | Cat: #12262 RRID: AB_2566811 | WB (1:5000) |
| Antibody | anti-human CD36 (rabbit polyclonal) | Thermo Fisher Scientific | Cat: # PA5-81996 RRID: AB_2789157 | Immunofluorescence (1:100) |
| Antibody | anti-human Mertk (rabbit monoclonal) | Thermo Fisher Scientific | Cat: # MA5-31991 RRID: AB_2809285 | Immunofluorescence (1:150) |
| Antibody | anti-human cytokeratin 18 (rabbit polyconal) | GeneTex | Cat: #GTX112977 RRID: AB_10719880 | Immunofluorescence (1:500) |

*Continued on next page*

*Continued*

| Reagent type (species) or resource | Designation | Source or reference | Identifiers | Additional information |
|---|---|---|---|---|
| Antibody | anti-dsRNA J2 (mouse monoclonal) | SCICONS English & Scientific Consulting Kft | Cat: #10010200 RRID: AB_2651015 | Immunofluorescence (1:1000) |
| Antibody | donkey anti-rabbit IgG HRP (donkey polyclonal) | Jackson ImmunoResearch | Cat: #711-036-152 RRID: AB_2340590 | WB (1:5000) |
| Antibody | Alexa-594 goat-anti mouse IgG (goat polyclonal) | Thermo Fisher Scientific | Cat: #A11005 RRID: AB_2534073 | Immunofluorescence (1:500) |
| Antibody | Alexa-488 donkey-anti rabbit IgG (donkey polyclonal) | Thermo Fisher Scientific | Cat: #A21206 RRID: AB_2535792 | FACS (1:2500) and Immunofluorescence (1:500) |
| Antibody | Alexa-594 goat-anti rabbit IgG (goat polyclonal) | Thermo Fisher Scientific | Cat: #A11012 RRID: AB_141359 | Immunofluorescence (1:500 or 1:1000) |
| Antibody | Alexa-488 F(ab)'2 goat-anti mouse IgG (goat polyclonal) | Thermo Fisher Scientific | Cat: #A11017 RRID: AB_2534084 | Immunofluorescence (1:500) |
| Antibody | Alexa-647 donkey-anti mouse IgG (donkey polyclonal) | Thermo Fisher Scientific | Cat: #A31571 RRID: AB_162542 | Immunofluorescence (1:500 or 1:1000) |
| Antibody | Alexa-647 goat anti-rabbit IgG (goat polyclonal) | Thermo Fisher Scientific | Cat: #A21244 RRID: AB_2535812 | Immunofluorescence (1:500) |
| Cell line (*Homo sapiens*) | Calu-3 | BCRJ #0264 | ATCC #HTB-55 | |
| Cell line (*H. sapiens*) | Jurkat E6-1 | BCRJ #0125 | ATCC #TIB-152 | |
| Cell line (*H. sapiens*) | THP-1 | Dario Zamboni's lab | ATCC: #TIB-202 | |
| Cell line (*Cercopithecus aethiops*) | Vero CCL81 | Dario Zamboni's lab | ATCC: #CCL-81 | |
| Strain, strain background (human) | SARS-CoV-2 | Brazil/SPBR-02/2020 | | |
| Strain, strain background (human) | Coxsackievirus B5 | Roger M. Loria's lab | | |
| Chemical compound, drug | FITC Annexin V | BD Biosciences | Cat: #556,420 | FACS (1:200) |
| Chemical compound, drug | Purified Recombinant Annexin V | BD Biosciences | Cat: #556,416 | |
| Chemical compound, drug | Zombie NIR Fixable Viability | Biolegend | Cat: #423,106 | FACS (1:400) |
| Chemical compound, drug | Zombie Violet Fixable Viability Kit | Biolegend | Cat: #423,114 | FACS (1:400) |
| Chemical compound, drug | Remdesivir | BOC Sciences | Cat: #1809249-37-3 | |
| Chemical compound, drug | phorbol 12-myristate 13-acetate (PMA) | InvivoGen | Cat: #Tlrl-pma | |
| Chemical compound, drug | Annexin V MicroBeads | Miltenyi Biotec | Cat: #130-090-201 | |
| Chemical compound, drug | Z-VAD-FMK | Selleckchem | Cat: #S7023 | |
| Chemical compound, drug | Glycine | Sigma | Cat: #G7126 | |
| Chemical compound, drug | complete, Mini Protease Inhibitor Cocktail | Sigma | Cat: #11836153001 | |
| Chemical compound, drug | Cytochalasin D | Sigma | Cat: #C8273 | |
| Chemical compound, drug | CellTrace CFSE Cell Proliferation Kit | Thermo Fisher Scientific | Cat: #C34554 | |
| Chemical compound, drug | CellTrace Far Red Cell Proliferation Kit | Thermo Fisher Scientific | Cat: #C34564 | |
| Chemical compound, drug | CellTrace Violet Cell Proliferation Kit | Thermo Fisher Scientific | Cat: #C34557 | |
| Chemical compound, drug | pHrodo Red AM Intracellular pH Indicator | Thermo Fisher Scientific | Cat: #P35372 | |
| Chemical compound, drug | Hoechst 33,342 | Thermo Fisher Scientific | Cat: #H3570 | |
| Commercial assay or kit | BD Cytofix/Cytoperm Fixation/Permeabilization Kit | BD Biosciences | Cat: #554,715 | |
| Commercial assay or kit | CytoTox 96 Non-Radioactive Cytotoxicity Assay | BD Biosciences | Cat: #G1780 | |

*Continued on next page*

*Continued*

| Reagent type (species) or resource | Designation | Source or reference | Identifiers | Additional information |
|---|---|---|---|---|
| Commercial assay or kit | Human Inflammatory Cytokine Cytometric Bead Array (CBA) | BD Biosciences | Cat: #551,811 | |
| Commercial assay or kit | Human IL-6 DuoSet ELISA | R&D Systems | Cat: #DY206-05 | |
| Commercial assay or kit | PowerUp SYBR Green Master Mix | Thermo Fisher Scientific | Cat: #A25742 | |
| Commercial assay or kit | Zenon Rabbit IgG Labeling Kits | Thermo Fisher Scientific | Cat: #Z25306 | |
| Commercial assay or kit | Direct-zol RNA Miniprep Kits | Zymo Research | Cat: #R2052 | |
| Software, algorithm | Adobe Photoshop | Adobe Inc | | |
| Software, algorithm | FlowJo v10.8.0 | FlowJo, LLC | | |
| Software, algorithm | Graphpad PRISM 8.0 | GraphPad Software, Inc | | |
| Software, algorithm | ImageJ (Fiji) | National Institutes of Health | | |
| Software, algorithm | NIS Elements | Nikon Instruments Inc. | | |
| Software, algorithm | ClusterProfiler | *Wu et al., 2021* | | |
| Software, algorithm | fgsea R | *Korotkevich et al., 2016* | | |
| Software, algorithm | R v4.0.4 | | | |
| Software, algorithm | Seurat v4.1.0 | *Stuart et al., 2019* | | |
| Software, algorithm | Enrichr | *Chen et al., 2013* | | |

## Study approval

The procedures followed in the study were approved by the Research Ethics Committee of Hospital das Clínicas de Ribeirão Preto (CEP-FMRP/USP) and by the National Ethics Committee, Brazil Comissão Nacional de Ética em Pesquisa, protocols 30248420.9.0000.5440 and 39722020.9.0000.5440. Written informed consent was obtained from recruited donors.

## Isolation of human peripheral blood mononuclear cells, monocyte seeding, and macrophage differentiation

Peripheral blood mononuclear cells (PBMC) from healthy donors were isolated from peripheral blood. The samples were centrifuged at 400 g for 10 min at room temperature (RT), the plasma was discarded, and the cellular portion was diluted in phosphate-buffered saline (PBS). PBMC were isolated by density–gradient centrifugation using Histopaque-1077 (Sigma-Aldrich). The collected PBMC fraction was treated with ACK lysis buffer to lyse erythrocytes and washed twice with PBS. For macrophages differentiation, monocytes were cultivated for 7 days in RPMI 1640 media supplemented with penicillin (10,000 U/mL, GIBCO), streptomycin (10,000 µg/mL, GIBCO), 1% glutamine (GIBCO), and 10% heat-inactivated human serum (Sigma Aldrich) at 37°C in a 5% $CO_2$ atmosphere.

## Cell lines

Calu-3 and Jurkat cells were originally from (ATCC) and purchased from Banco de Células do Rio de Janeiro (BCRJ). Vero CCL81 and THP-1 cells were from Dario Zamboni's lab. No authentication method was used. Calu-3 and Vero CCL81 cells were maintained in Dulbecco's Modified Eagle Media (DMEM) with 4.5 g/L glucose (GIBCO) supplemented with penicillin (10,000 U/mL), streptomycin (10,000 µg/mL), and 10% heat-inactivated fetal bovine serum (FBS, GIBCO) (DMEMc) at 37°C in a 5% $CO_2$ atmosphere. Jurkat cells and THP-1 cells were maintained in RPMI1640 media supplemented with penicillin (10,000 U/mL), streptomycin (10,000 µg/mL), 1% glutamine, and 10% FBS (RPMIc) at 37°C in a 5% $CO_2$ atmosphere.

THP-1-derived macrophages were obtained by treatment with phorbol 12-myristate 13-acetate (PMA, 50 ng/mL, Sigma-Aldrich) for 24 hr, followed by media replenishment with fresh RPMIc and another 24 hr incubation period before stimulation.

Cell lines were routinely screened by PCR for mycoplasma contamination, and all cell lines tested negative.

## Viral stock

The SARS-CoV-2 strain (Brazil/SPBR-02/2020) was isolated from the first Brazilian case of COVID-19. The viral stock was produced in a monolayer of Vero CCL81 cells expressing ACE2 (*Rodrigues et al., 2021*), maintained in serum-free DMEM at 37°C in a 5% $CO_2$ atmosphere. Coxsackievirus B5 (Coxsackie) was originally shared by Dr Roger M Loria (Virginia Commonwealth University, Richmond, Virginia, USA) (*Gomes et al., 2010*). The viral stock was produced in a monolayer of HeLa cells maintained in DMEM supplemented with 2% FBS at 37°C and 5% $CO_2$. Viral stocks were propagated under BSL3 conditions and stored at –80°C. Viral loads were estimated by titration in Vero CCL81 cells seeded onto 96-well plates and standard limiting dilution to confirm the 50% tissue culture infectious dose.

## Infection and UV irradiation of epithelial cell lines

Cell monolayers were rinsed with PBS, replenished with a thin layer of serum-free media, and infection with a freshly thawed virus aliquot at a multiplicity of infection (MOI) of approximately 0.05 was carried out for 1 hr at 37°C and 5% $CO_2$ for viral adsorption. The cells were then topped with fresh media and incubated for 48 hr. Sterile AC (UV-AC) were generated by exposure to UV-C radiation (Calu-3 cells at 500 mJ/cm$^2$ and Vero CCL81 cells at 350 mJ/cm$^2$) in a UV crosslinker (Fisher Scientific), followed by 6 hr incubation in DMEMc at 37°C and 5% $CO_2$.

## Cell death assays

Cells were collected 48 hr post-infection or 6 hr post-irradiation, spun at 300 g for 5 min, washed with PBS, and resuspended in the appropriate buffer before analysis.

For the analysis of caspase-8 and caspase-3 cleavage, $2.0 \times 10^6$ Vero CCL81 and Calu-3 cells stimulated as described were lysed in radio immuno-precipitation assay (RIPA) buffer (10 mM Tris-HCl, pH 7.4, 1 mM EDTA, 150 mM NaCl, 1% Nonidet P-40, 1% deoxycholate, and 0.1% SDS) supplemented with protease inhibitor cocktail (Sigma). Precleared lysates were quantified by Bradford assay, boiled in Laemmli buffer, resolved by SDS-PAGE, and transferred to poly(vinylidene fluoride) membranes. Rabbit anti-cleaved caspase-3/caspase-3/cleaved caspase-8/caspase-8 (all from Cell Signaling), rabbit anti-SARS-CoV-2 Spike (Abcam), HRP-conjugated goat anti-actin (Cell Signaling), and species-specific horseradish-conjugated secondary antibodies were used for antigen detection.

For the analysis of cleaved caspase-3 by flow cytometry, infection of Calu-3 cells was carried out in the presence of Z-VAD-FMK (Selleckchem) at 20 µM, or mock treatment with vehicle (DMSO), added to cell cultures after virus adsorption. $10^6$ collected cells were washed with PBS, and fixation, permeabilization, and staining were carried out using BD Cytofix/Cytoperm Fixation/Permeabilization Kit (BD Biosciences), following the manufacturer's instructions. Samples were incubated with rat serum (1:400) for 5 min at RT followed and immunolabeling of intracellular active caspase-3 was performed with anti-cleaved caspase-3 (Cell Signaling, 1:400) and secondary donkey anti-rabbit IgG conjugated to Alexa Fluor 488 (Thermo Fisher Scientific, 1:2,500). Sample acquisitions were performed in a FACSVerse (BD Biosciences) flow cytometer and data processing was performed using FlowJo_V10 software (BD Biosciences).

Cell death profiling in infected or UV-irradiated Vero CCL81 cells was performed with a viability dye (Zombie NIR Fixable Viability Dye, Biolegend, 1:400) and annexin V (FITC annexin V, BD Biosciences, 1:200) co-labeling. Labeling of $1.0 \times 10^6$ cells with Zombie NIR was carried out for 10 min on ice, followed by annexin V labeling for 15 min on ice with cells resuspended annexin V binding buffer (0.01 M HEPES, 0.14 M NaCl, 2.5 mM $CaCl_2$, pH 7.4). Sample acquisitions were performed in a BD FACSMelody (BD Biosciences) cell sorter and data processing was performed using FlowJo_V10 software (BD Biosciences).

To determine LDH release, infection or UV irradiation of $2.0 \times 10^5$ Vero CCL18 cells was carried out in DMEMc without Phenol Red. LDH release was measured in the culture supernatants using the CytoTox 96 non-radioactive cytotoxicity assay (Promega), following the manufacturer's instructions.

## Isolation of apoptotic cells

AC were generated from SARS-CoV-2-infected (CoV2-AC) or UV-irradiated (UV-AC) Vero CCL81 and Calu-3 cells, or Coxsackievirus B5-infected Vero CCL81 cells (Coxsackie-AC). At the established time points, the supernatants were collected and temporarily stored for further processing. Then, a

suspension containing CoV2-AC, Coxsackie-AC, or UV-AC was obtained by carefully pipetting fresh media up and down the plate surface and collecting loosely attached cells. Following spun at 300 g for 5 min, pelleted AC were washed with PBS, resuspended in RPMIc, and counted prior to incubation with macrophages. Enrichment in AC was confirmed by annexin V staining and flow cytometry.

To sort CoV2-AC populations, annexin $V^+$ cells were pre-enriched using magnetic separation (Annexin V MicroBead Kit, Miltenyi Biotec) following the manufacturer's instructions. Then cells were labeled with Zombie NIR and annexin V-FITC, as described above, and cell sorting was performed in a BD FACSMelody (BD Biosciences) cell sorter. Isolated CoV2-AC were washed in PBS, resuspended in fresh RPMIc, and counted prior to incubation with macrophages.

For virus inactivation, isolated CoV2-AC suspension were transferred to a culture dished and placed onto tissue culture hood surface with an open lid for 20 min with the UV-C lamp on. Virus inactivation was confirmed by $TCID_{50}$, as described below. Alternatively, isolated AC were fixed with 2% PFA in PBS for 10 min and washed twice with PBS before adding to macrophage cultures.

To prepare the cell-free conditioned supernatants of the infected AC (CoV2-AC Sup), collected supernatants were spun at 300 g for 5 min to remove cellular debris; the resulting supernatant was filtered using a 0.22 μm syringe filter and immediately used for macrophage stimulation. The volume of CoV2-AC Sup used to stimulate macrophages was normalized to the amount of collected AC.

## $TCID_{50}$ quantification of viable viral particles

$1.0 \times 10^6$ isolated CoV2-AC and their equivalent purified conditioned supernatants, obtained as described above, were lysed by snap freeze-and-thaw. To estimate viral load in AC isolated by annexin V labeling, all infected cells were collected, followed by magnetic separation with annexin V microbeads (Miltenyi Biotec) according to the manufacturer's protocol. $1.0 \times 10^6$ pre-sorted cells and annexin$^+$ cells were lysed by freeze-and-thaw for quantification. Viral loads were estimated by titration in Vero CCL81 cells seeded onto 96-well plates and expressed as 50% tissue culture infectious dose ($TCID_{50}$). Quantification was performed with the Reed-Muench method and plotted as $TCID_{50}$ units per mL (*Reed and Muench, 1938*).

## In vitro assays with THP-1- and PBMC-derived macrophages

$1.0 \times 10^6$ THP-1- and PBMC-derived macrophages seeded on 24-well tissue culture plates were rinsed with warm PBS before stimulation. Macrophage cultures were then topped with UV-AC, CoV2-AC, or Coxsackie-AC resuspended in the appropriate serum-containing macrophage media at a 1:1 ratio. Macrophage cultures were rinsed out 2 hr after incubation to remove unbound AC and topped with fresh media. At the indicated time points, cell supernatants were collected, and macrophages were rinsed at least twice with warm PBS to remove cell debris before proceeding to further analysis. To block AC ligation and efferocytosis, CoV2-AC were pre-incubated with 0.1 or 0.01 μg/mL of annexin V (BD Biosciences), and macrophage stimulation was carried out in RPMIc supplemented with $Ca^{2+}$. To block AC internalization, macrophages were pre-treated for 30 min, and incubation was carried out in the presence of Cytochalasin D (10 μM, Sigma). Where indicated, macrophages were topped with fresh media containing Remdesivir (20 μM, BOC Sciences) after removal of unbound AC.

For stimulation with conditioned supernatants, CoV2-AC-Sup were added directly to macrophage cultures and topped with fresh media for volume adjustment. Infection with SARS-CoV-2 or Coxsackievirus was carried out with virus adsorption at MOI 0.1 in serum-free media for 1 hr before topping with fresh serum-containing media.

For gene expression analysis by RT-qPCR, rinsed macrophage cultures were lysed in Trizol (Thermo Scientific) and immediately frozen at –80°C.

To estimate CD206 surface expression, rinsed macrophages were removed with a cell scraper, spun at 300 g for 5 min, washed with PBS, followed by incubation with rat serum (1:400) for 5 min at RT and cell surface staining for 15 min at 4°C. Cells were labeled with Zombie Violet Fixable Viability Kit (Biolegend) and anti-CD206-FITC antibody (clone 19.2, BD Biosciences). The cells were then fixed with 2% PFA for 10 min and washed before flow cytometric analyses in a FACSVerse (BD Biosciences) flow cytometer and data processing was performed using FlowJo_V10 software (BD Biosciences).

For immunofluorescence imaging of efferocytosis in vitro, human monocyte-derived macrophages seeded onto 13 mm round coverslips were incubated with UV-AC or CoV2-AC labeled with 5 μM CellTrace CFSE dye (Thermo Scientific) for 2 hr. After rinsing out non-engulfed AC, macrophages

were fixed for 20 min with PBS 2% PFA at RT. Cells were then permeabilized with 0.1% Triton-X-100 in PBS for 10 min at RT. Next, cells were rinsed twice in PBS and incubated with PBS 1% bovine serum albumin (BSA) and 5 µg/mL normal donkey IgG (Jackson ImmunoResearch) for 45 min RT prior to overnight (ON) incubation at 4°C with primary antibodies: rabbit polyclonal anti-SARS-CoV-2 Spike antibody (Abcam, 1:1000) and mouse mAb anti-rat CD11b (Abcam, 1:300) diluted in PBS containing 1% BSA. Next, cells were rinsed thoroughly in PBS and incubated for 30 min at RT with the secondary antibodies diluted in PBS: Alexa-594 goat-anti rabbit IgG (1:1000; Thermo Scientific) and Alexa-647 donkey-anti mouse IgG (1:1000; Thermo Scientific). Cell nuclei were stained with Hoechst 33,342. Cells were then rinsed, and coverslips mounted on glass slides with Fluoromount-G (Invitrogen). Samples were imaged using a Zeiss LSM 780 laser scanning confocal microscope (Carl Zeiss), post-processed for brightness and contrast adjustment with ImageJ image processing package (NIH), and layouts were built on Adobe Photoshop.

For the engulfment assays by flow cytometry, macrophages were labeled with 5 µM CellTrace Violet dye (CTV, Thermo Scientific), according to the manufacturer's recommendations. CellTrace Violet (CTV)-labeled macrophages were stimulated with UV-AC or CoV2-AC previously stained with 1 µM CellTrace Far Red dye (CTFR, Thermo Scientific) or 5 µM CellTrace CFSE dye, as described in the legend of the figures. Following the removal of unbound AC, macrophages were collected with a cell scraper, fixed with 2% PFA for 10 min, and washed before proceeding with the acquisition by flow cytometry. For the two-round efferocytosis assay, first-round UV-AC and CoV2-AC were labeled with a pH indicator fluorogenic intracellular probe (pHRodo Red AM Intracellular pH Indicator - Thermo Scientific). Second-round apoptotic Jurkat cells (UV-Jurkat) were generated by labeling with 1 µM CTFR and UV-C irradiation at 20 mJ/cm$^2$, followed by incubation at 37°C and 5% $CO_2$ for 6 hr. CTV-labeled THP-1 cells were incubated with pHrodo-UV-AC or pHrodo-CoV2-AC for 18 hr at a 1:1 ratio. Cells were then rinsed with PBS, and CTV-labeled UV-Jurkat cells were added to the culture at a 1:1 ratio for 2 hr. Macrophage cultures were collected with a cell scraper, washed, fixed with 2% PFA for 10 min, and washed before proceeding with the acquisition by flow cytometry. Flow cytometric acquisition was performed in a FACSVerse (BD Biosciences) flow cytometer and data processing was performed using FlowJo_V10 software (BD Biosciences).

## Cytokine quantification

IL-6 and IL-1β cytokine levels in macrophage supernatants were evaluated by ELISA assay (R&D Systems) or cytometric bead array (CBA, BD Biosciences), as indicated in the legend of the figures, following the manufacturers' recommendations.

## RNA isolation and gene expression analyses

Total RNA extraction for human monocyte- or THP-1-derived macrophages stimulated in vitro was performed using Directzol RNA miniPrep Kit (Zymo Research), following the manufacturer's recommendations. Reverse transcriptase was performed using Moloney Murine Leukemia Virus reverse transcriptase (Thermo Scientific) according to the manufacturer's recommendations.

Gene expression RT-qPCR was performed using SybrGreen Master Mix (Applied Biosystems). RT-qPCR was performed in fast mode, following the manufacturer's recommendations. The evaluation of each gene expression was determined by the comparative CT method. Primer sequences used were:

| Gene | 5' Forward | 3' Reverse |
| --- | --- | --- |
| CCL18 | CTCTGCTGCCTCGTCTATACCT | CTTGGTTAGGAGGATGACACCT |
| CD163 | GCGGGAGAGTGGAAGTGAAAG | GTTACAAATCACAGAGACCGCT |
| CD36 | GGGAAAGTCACTGCGACATG | TGCAATACCTGGCTTTTCTCA |
| GAPDH | GTCTCCTCTGACTTCAACAGCG | ACCACCCTGTTGCTGTAGCCAA |
| IL6 | GGTACATCCTCGACGGCATCT | GTGCCTCTTTGCTGCTTTCAC |
| ITGB5 | AACCAGAGCGTGTACCAGAA | AGGAGAAGTTGTCGCACTCA |
| MERTK | CTCTGGCGTAGAGCTATCACT | AGGCTGGGTTGGTGAAAACA |

*Continued on next page*

*Continued*

| Gene | 5' Forward | 3' Reverse |
|------|-----------|-----------|
| MMP9 | TGTACCGCTATGGTTACACTCG | GGCAGGGACAGTTGCTTCT |
| MRC1 | AGCCAACACCAGCTCCTCAAGA | CAAAACGCTCGCGCATTGTCCA |
| PPARG | ACCAAAGTGCAATCAAAGTGGA | ATGAGGGAGTTGGAAGGCTCT |
| SRA1 | GCAGTGGGATCACTTTCACAA | AGCTGTCATTGAGCGAGCATC |
| TIM4 | ACAGGACAGATGGATGGAATACCC | AGCCTTGTGTGTTTCTGCG |

## Histological analysis

Ultrasound-guided minimally invasive autopsies for COVID-19 deceased patients were approved by the Research Ethics Committee of Hospital das Clínicas de Ribeirão Preto (CEP, protocol no. 4.089.567). *Supplementary file 2* describes clinical, laboratory, and treatment records of COVID-19 patients. Non-neoplasic sections of lung parenchyma were obtained from lobectomies for lung cancer as a control group (*Rodrigues et al., 2021*). Paraffin sections from lungs were collected and processed as previously described (*Rodrigues et al., 2021*). Sections were rinsed in PBS, and the antigen retrieval was performed by incubation of samples with 0.1% trypsin (ThermoFisher Scientific) in PBS at 37°C for 15 min. Next, sections were rinsed and incubated with Image-iT FX Signal Enhancer (ThermoFisher Scientific) for 30 min at RT. Sections were then rinsed in PBS and incubated for 45 min at RT in PBS containing 0.5% BSA and 5 μg/mL normal donkey IgG (Jackson ImmunoResearch).

For analysis of active caspase-3, lung sections were labeled with primary antibodies diluted in PBS and incubated overnight at 4°C: rabbit polyclonal anti-human cleaved caspase-3 (Asp175) (Cell Signaling; 1:1000) and mouse anti-dsRNA (SCICONS; 1:1000). Then, sections were rinsed thoroughly in PBS and incubated for 45 min at RT with the secondary antibodies: Alexa 594 goat anti-rabbit IgG (1:500; Thermo Fisher Scientific) and Alexa 488 goat anti-mouse IgG F(ab)´₂ (1:500; Thermo Fisher Scientific) diluted in PBS. Cell nuclei were stained with Hoechst 33,342. Cells were then rinsed, and coverslips mounted on glass slides with Fluoromount-G (Invitrogen). For in situ efferocytosis analysis, lung sections were labeled with primary antibodies mouse mAb anti-human CD68 (Abcam; 1:100), rabbit mAb anti-human cytokeratin 18 (GeneTex, 1:500) diluted in PBS and incubated overnight at 4°C. Then, sections were rinsed thoroughly in PBS and incubated for 45 min at RT with the secondary antibodies F(ab)'₂ goat anti-mouse IgG conjugated to Alexa 488 (1:500; Thermo Scientific) and goat anti-rabbit IgG conjugated to Alexa 647 (1:500; Thermo Scientific). Next, sections were rinsed in PBS and incubated overnight at 4°C with rabbit polyclonal anti-SARS-CoV-2 Spike (Abcam; 1:1000) previously conjugated to Alexa 568, according to the manufacturer's instructions (Zenon Alexa Fluor 568 Rabbit IgG Labeling Kit, Thermo Scientific). Sections were then rinsed in PBS and incubated for 15 min at RT with Hoechst 33,342 (Invitrogen), rinsed thoroughly, and mounted on glass slides with Fluoromount-G (Invitrogen). Samples from Cl-caspase-3 and in situ efferocytosis experiments were imaged using an ECLIPSE Ti2 Series microscope equipped with a DS-Qi2 camera (Nikon Instruments Inc). Analysis was performed using NIS Elements (Nikon Instruments Inc), wherein mean intensity fluorescence for Cl-caspase-3 was determined for randomly selected epithelial cells. Representative images were postprocessed for brightness and contrast adjustment with ImageJ image processing package (NIH) and layouts were built on Adobe Photoshop.

For CD36 or MERTK expression analysis, lung sections were labeled with the primary antibodies diluted in PBS and incubated overnight at 4°C: rabbit polyclonal anti-human CD36 (PA5-81996; 1:100 ThermoFisher Scientific) or rabbit mAb anti-human MERTK (Clone SR29-07; 1:150; ThermoFisher Scientific) and mouse mAb anti-human calprotectin (Clone MAC387; MA1-81381; 1:100; ThermoFisher Scientific). Then, sections were rinsed thoroughly in PBS and incubated for 45 min at RT with the secondary antibodies Alexa 488 donkey anti-rabbit IgG (1:500; Thermo Fisher Scientific) and Alexa 594 goat anti-mouse IgG (Thermo Scientific) diluted in PBS. Sections were then rinsed, and coverslips mounted with DAPI Fluoromount-G. Samples were imaged using an Olympus BX61 Fluorescence Motorized Slide Scanner Microscope Pred VS120 (Olympus). Analysis was performed using ImageJ image processing package (NIH), wherein mean intensity fluorescence for CD36 and MERTK

was determined for randomly selected S1009[+] cells. Representative images were postprocessed for brightness and contrast adjustment with ImageJ image processing package (NIH) and layouts were built on Adobe Photoshop.

### Re-analysis of scRNA-seq data

We re-analyzed single-cell transcriptomic data from BAL fluid from patients with varying severity of COVID-19 disease and their respective healthy controls (*Liao et al., 2020*). The dataset is publicly available at https://covid19-balf.cells.ucsc.edu/. Downloaded data was imported into R environment version v3.6.3. Differential expression analysis was conducted using FindMarkers function in Seurat using the Wilcoxon test to compare mild and severe COVID-19 patients with healthy individuals for each cluster previously identified by authors. Differentially expressed genes between mild/severe COVID-19 patients and controls for each cluster were identified considering genes expressed in at least 5% of cells and FDR <0.05 and |avg_logFC|>0.1. In addition, a differentially expressed gene list comparing infected (identified SARS-Cov-2 transcripts) against non-infected alveolar macrophages was obtained from supplemental materials from *Ren et al., 2021*. Gene set enrichment analysis (GSEA) was performed using ClusterProfiler and fgsea R packages (*Korotkevich et al., 2016*; *Wu et al., 2021*) for each differentially expressed gene or log2 fold change lists from *Liao et al., 2020* and *Ren et al., 2021* dataset. A custom gene set was also utilized as a reference for GSEA, incorporating GO:0006911, GO:0043277 GO terms, and additional efferocytic related genes from the literature (*Penberthy and Ravichandran, 2016*; *Boada-Romero et al., 2020*) (*Supplementary file 1*). Significantly enriched sets were identified considering terms with p-value <0.05.

### Quantification and statistical analyses

Please refer to the legend of the figures for description of samples, sample sizes (biological replicates), and experimental replicates. No statistical tests were used to estimate sample size. Data were plotted and analyzed with GraphPad Prism 8.4.2 software (GraphPad Prism Software Inc, San Diego, CA). The statistical tests used are listed in the legend of the figures. For the in vitro assays, we used Student's t-test to compare two experimental groups or ANOVA and multiple comparison correction to compare three or more experimental groups. For quantification of histological datasets, normality tests were performed, and samples with non-Gaussian distribution were analyzed applying the Mann–Whitney test.

## Acknowledgements

The authors thank Dr Denise M da Fonseca, Msc Elizabete R Milani and Dr Roberta R C Rosales (FMRP-USP) for technical assistance. Funding: This work was supported by grants from Fundação de Amparo a Pesquisa do Estado de São Paulo (FAPESP) grants 2018/25559–4 and 2020/05288–6 (LDC); Coordenação de Aperfeiçoamento de Pessoal de Nível Superior (CAPES) grant 88887.507253/2020–00 (D S Z and LDC.); Conselho Nacional de Desenvolvimento Científico e Tecnológico (CNPq) grant 434538/2018–3 (LDC); D S, A C G S, T S R, M F R, E G F F, D L A T are supported by FAPESP fellowships.

## Additional information

### Funding

| Funder | Grant reference number | Author |
| --- | --- | --- |
| Fundação de Amparo à Pesquisa do Estado de São Paulo | 2018/25559-4 | Larissa D Cunha |
| Fundação de Amparo à Pesquisa do Estado de São Paulo | 2020/05288-6 | Larissa D Cunha |
| Coordenação de Aperfeiçoamento de Pessoal de Nível Superior | 88887.507253/2020-00 | Dario S Zamboni |

| Funder | Grant reference number | Author |
| --- | --- | --- |
| Conselho Nacional de Desenvolvimento Científico e Tecnológico | 434538/2018-3 | Larissa D Cunha |

The funders had no role in study design, data collection and interpretation, or the decision to submit the work for publication.

## Author contributions

Ana CG Salina, Conceptualization, Data curation, Formal analysis, Investigation, Methodology, Validation, Visualization, Writing – original draft, Writing – review and editing; Douglas dos-Santos, Writing – review and editing, Conceptualization, Formal analysis, Investigation, Methodology, Validation, Writing – original draft; Tamara S Rodrigues, Conceptualization, Formal analysis, Investigation, Methodology, Writing – original draft; Marlon Fortes-Rocha, Edismauro G Freitas-Filho, Conceptualization, Formal analysis, Investigation, Writing – original draft; Daniel L Alzamora-Terrel, Conceptualization, Formal analysis, Writing – original draft; Icaro MS Castro, Formal analysis, Validation, Writing – original draft; Thais FC Fraga da Silva, Leticia Almeida, Adriene Y Ishimoto, Formal analysis, Writing – original draft; Mikhael HF de Lima, Daniele C Nascimento, Camila M Silva, Juliana E Toller-Kawahisa, Amanda Becerra, Samuel Oliveira, Diego B Caetité, Thais M Lima, Ronaldo B Martins, Maira N Benatti, Sabrina S Batah, Eurico Arruda Neto, Thiago M Cunha, José C Alves-Filho, Vania LD Bonato, Fernando Q Cunha, Dario S Zamboni, Paulo Louzada-Junior, Resources, Writing – original draft; Flavio Veras, Alexandre T Fabro, Rene DR Oliveira, Writing – review and editing, Resources, Writing – original draft; Natália B do Amaral, Marcela C Giannini, Letícia P Bonjorno, Maria IF Lopes, Rodrigo C Santana, Fernando C Vilar, Maria A Martins, Rodrigo L Assad, Sergio CL de Almeida, Fabiola R de Oliveira, Writing – review and editing, Writing – original draft; Helder I Nakaya, Conceptualization, Resources, Writing – original draft; Larissa D Cunha, Data curation, Funding acquisition, Investigation, Project administration, Resources, Supervision, Visualization, Writing – original draft

## Author ORCIDs

Ana CG Salina http://orcid.org/0000-0003-3220-0413
Douglas dos-Santos http://orcid.org/0000-0002-7155-0443
Marlon Fortes-Rocha http://orcid.org/0000-0002-2055-2000
Edismauro G Freitas-Filho http://orcid.org/0000-0002-1910-1085
Thais M Lima http://orcid.org/0000-0002-5714-6057
Rodrigo L Assad http://orcid.org/0000-0002-8430-8357
Thiago M Cunha http://orcid.org/0000-0003-1084-0065
Helder I Nakaya http://orcid.org/0000-0001-5297-9108
Dario S Zamboni http://orcid.org/0000-0002-7856-7512
Larissa D Cunha http://orcid.org/0000-0002-1290-0263

## Ethics

The procedures followed in the study were approved by the Research Ethics Committee of Hospital das Clínicas de Ribeirão Preto (CEP-FMRP/USP) and by the National Ethics Committee, Brazil Comissão Nacional de Ética em Pesquisa (CONEP), protocols 30248420.9.0000.5440 and 39722020.9.0000.5440. Written informed consent was obtained from recruited donors. Ultrasound-guided minimally invasive autopsies for COVID-19 deceased patients were approved by the Research Ethics Committee of Hospital das Clínicas de Ribeirão Preto (CEP, protocol no. 4.089.567).

## Decision letter and Author response

Decision letter https://doi.org/10.7554/eLife.74443.sa1
Author response https://doi.org/10.7554/eLife.74443.sa2

# Additional files

## Supplementary files

• Supplementary file 1. Custom gene sets incorporating associated to efferocytic pathway and related to human diseases (related to *Figure 5C*).

• Supplementary file 2. Medical characteristics of COVID-19 patients.

• Transparent reporting form

## Data availability

Source data files containing the numerical values for graphs depicting flow cytometry, ELISA, CBA, RT-qPCR, and imaging quantification data have been uploaded as csv files.

The following previously published datasets were used:

| Author(s) | Year | Dataset title | Dataset URL | Database and Identifier |
|---|---|---|---|---|
| Liao M, Liu Y, Yuan J, Wen Y, Xu G, Zhao J, Cheng L, Li J, Wang X, Wang F, Liu L, Amit I, Zhang S, Zhang Z | 2020 | Single-cell landscape of bronchoalveolar immune cells in patients with COVID-19 | https://www.ncbi.nlm.nih.gov/geo/query/acc.cgi?acc=GSE145926 | NCBI Gene Expression Omnibus, GSE145926 |
| Ren X, Wen W, Fan X, Hou W, Bin Su, Cai P, Li J, Liu Y, Tang F, Zhang F, Yang Y, Jiangping He, Ma W, Jingjing He, Wang P, Cao Q, Chen F, Chen Y, Cheng X, Deng G, Deng X, Ding W, Feng Y, Gan R, Guo C, Guo W, He S, Jiang C, Liang J, Li Y, Lin J, Ling Y, Liu H, Liu J, Liu N, Liu S-Q, Luo M, Ma Q, Song Q, Sun W, Wang G, Wang F, Wang Y, Wen X, Wu Q, Xu G, Xie X, Xiong X, Xing X, Xu H, Yin C, Yu D, Yu K, Yuan J, Zhang B, Zhang P, Zhang T, Zhao J, Peidong Zhao, Zhou J, Zhou W, Zhong S, Zhong X, Zhang S, Zhu L, Zhu P, Zou B, Zou J, Zuo Z, Bai F, Huang X, Zhou P, Jiang Q, Huang Z, Bei J-X, Wei L, Bian X-W, Liu X, Cheng T, Li X, Wang F-S, Wang H, Bing Su, Zheng Z, Qu K, Wang X, Chen J, Jin R, Zemin Z, Zhao P | 2021 | COVID-19 immune features revealed by a large-scale single-cell transcriptome atlas | https://www.ncbi.nlm.nih.gov/geo/query/acc.cgi?acc=GSE158055 | NCBI Gene Expression Omnibus, GSE158055 |

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
