## [Editor Report]

The lung and other organ damage sustained during Covid-19 disease results predominantly from the immune response to SARS-CoV-2, which includes inflammation. In this paper the authors investigated potential mechanisms, and determined that part of the reason is the interaction of SARS-CoV-2 with macrophages, the cells of the innate immune response which have multiple roles, including efferocytosis, which is the engulfment of apoptotic cells, removing these cells and their debris. They found that SARS-CoV-2 virus is still viable in the apoptotic cells, and leads to an increase in macrophage production of inflammatory cytokines, as well as decreases their efferocytosis capability. In combination, these effects may lead to a more dysregulated immune response to the virus.

---

## [Decision Letter]

**Decision letter after peer review:**

Thank you for submitting your article "Efferocytosis of SARS-CoV-2-infected dying cells impairs macrophage anti-inflammatory functions and clearance of apoptotic cells" for consideration by *eLife*. Your article has been reviewed by 2 peer reviewers, and the evaluation has been overseen by a Reviewing Editor and Carla Rothlin as the Senior Editor. The following individual involved in review of your submission has agreed to reveal their identity: Rio Sugimura (Reviewer #2).

Essential revisions:

In this paper, the authors demonstrate that the macrophage efferocytic response in SARS-CoV-2 infection is compromised.

While these results are important, the following needs to be addressed:

1) The authors use Annexin V only to determine if a cell is apoptotic. This is misleading, as while Annexin V recognizes flipped phosphatidylserine on the surface of an apoptotic cell, if a cell is permeabilized (e.g. necrosis), it will also enter the cell and stain phosphatidylserine on the inside. Therefore, to conclusively determine cells are apoptotic, they should be double stained with a cell permeability dye such as propidium iodide (PI). An apoptotic cell will have an intact membrane (PI negative) and flipped phosphatidylserine (Annexin V positive). A double positive cell may have died by necrosis or other form of non-apoptotic cell death.

2) The results in circulating monocytes are inconsistent with the main mechanism presented in the paper. Either a mechanism linking the two results is presented or the circulating monocyte results should be removed.

3) Figure 1A-B show some examples of macrophages internalizing SARS-CoV-2 infected cells in a lung section from one deceased participant but no quantitation. Furthermore, the high background fluorescence in the Caspase 3 and dsRNA channels makes these panels confusing. Having these panels as the first presented results will decrease the impact of the paper.

*Reviewer #1 (Recommendations for the authors):*

This study, with revisions, makes interesting and important contributions to the COVID-19 literature. I question whether the patient monocyte data should be incorporated into this manuscript.

The authors need to fully characterize the necrotic cells present in "AC" fraction and to assess the relative roles of SARS-CoV-2 infected apoptotic and necrotic cells on in vitro macrophage behavior.

*Reviewer #2 (Recommendations for the authors):*

1. Could the anti-COVID19 pills ameliorate the efferocytosis of macrophages ingesting CoV-2 AC? The two of the most promising anti-COVID19 pills at this stage, Molnupiravir (Merck, available from Selleckchem for research purposes), and PF-07321332 (Pfizer, available from MedChemExpress for research purposes) could be tested. The experiments could be done in the same format as Figure 3B, two-round of efferocytosis assay using THP1.

2. What would be the effect of the COVID19 vaccine? Brazil reached over 55% or fully vaccinated population. Could the authors evaluate the expression of efferocytosis genes in PBMCs from healthy, COVID19 infected unvaccinated, COVID19 infected vaccinated? The same format of the experiment as Figure 4E.

---

## [Author Response]

While these results are important, the following needs to be addressed:1) The authors use Annexin V only to determine if a cell is apoptotic. This is misleading, as while Annexin V recognizes flipped phosphatidylserine on the surface of an apoptotic cell, if a cell is permeabilized (e.g. necrosis), it will also enter the cell and stain phosphatidylserine on the inside. Therefore, to conclusively determine cells are apoptotic, they should be double stained with a cell permeability dye such as propidium iodide (PI). An apoptotic cell will have an intact membrane (PI negative) and flipped phosphatidylserine (Annexin V positive). A double positive cell may have died by necrosis or other form of non-apoptotic cell death.

We thank the editor for this assessment. To address this concern, we performed costaining of cells with annexin V and a cell viability dye and evaluated them by flow cytometry. Our results show that, at 48h post-infection, most dying cells are apoptotic (Ann^+^ Zombie^-^), similar to UV-irradiate cells (Figure 1 C). To further probe the possible occurrence of necrosis (or non-apoptotic regulated cell death) in response to SARSCoV-2, we also performed a cytotoxicity assay. We confirmed that infection did not induce robust release of LDH, which would be expected in permeabilized cells (Figure 1D). We also highlight that our former data assessing apoptosis by flow cytometry of active caspase-3 (Figure 1B) has been complemented with immunoblot detection of cleaved caspase-3 and caspase-8 in infected cells (Figure 1A). Together, we believe the added data strongly supports the occurrence of apoptosis.

2) The results in circulating monocytes are inconsistent with the main mechanism presented in the paper. Either a mechanism linking the two results is presented or the circulating monocyte results should be removed.

We appreciate this assessment. Because understanding the interplay between local and systemic phenomena will take a considerable amount of work that surpasses the scope of this revision process, we decided to support the editor suggestion and limit the data presented in the current manuscript to those concerning a direct effect of efferocytosis of infected apoptotic cells. We are very interested in understanding how SARS-CoV-2 infection in the lungs affects efferocytic capacity systemically and expect to publish our findings in a follow-up study.

3) Figure 1A-B show some examples of macrophages internalizing SARS-CoV-2 infected cells in a lung section from one deceased participant but no quantitation. Furthermore, the high background fluorescence in the Caspase 3 and dsRNA channels makes these panels confusing. Having these panels as the first presented results will decrease the impact of the paper.

Following editorial advice, we moved the data on cleaved caspase-3 in situ to Figure 1E and present clearer, magnified image panels with improved background correction, accompanied by representative images of controls samples (obtained from benign lung tissue from adenocarcinoma biopsies) (Figure 1E) and COVID-19 samples stained with secondary antibodies only (Figure 1 —figure supplement 1A). Cleaved caspase-3 staining in lung section is now quantified in Figure E. While we are able to provide qualitative evidence of efferocytosis in situ in patient samples, we would not be able to quantify such an event meaningfully. But, because of the relevance of the qualitative finding to support our main hypothesis, we kept an improved version of the original data on Figure 1F.

Reviewer #1 (Recommendations for the authors):This study, with revisions, makes interesting and important contributions to the COVID-19 literature. I question whether the patient monocyte data should be incorporated into this manuscript.

We thank the reviewer for the positive assessment of our work.

The authors need to fully characterize the necrotic cells present in "AC" fraction and to assess the relative roles of SARS-CoV-2 infected apoptotic and necrotic cells on in vitro macrophage behavior.

We ask kindly that the reviewer refers to our response to major point #1 in the public review.

Reviewer #2 (Recommendations for the authors):1. Could the anti-COVID19 pills ameliorate the efferocytosis of macrophages ingesting CoV-2 AC? The two of the most promising anti-COVID19 pills at this stage, Molnupiravir (Merck, available from Selleckchem for research purposes), and PF-07321332 (Pfizer, available from MedChemExpress for research purposes) could be tested. The experiments could be done in the same format as Figure 3B, two-round of efferocytosis assay using THP1.

This is an interesting suggestion. Because of its commercial availability in Brazil at the time of this revision process, we chose Remdesivir, also approved to treat early-stage COVID-19, to test the effect of antiviral drugs targeting viral RNA transcription (RdRpdependent RNA synthesis). We found that Remdesivir treatment following stimulation of primary macrophages with CoV2-AC reduced IL-6 secretion, suggesting that sgRNA expression is important to modulate the inflammatory response (Figure 3H). We also examined Remdesivir effect in the two-round of efferocytosis assay and found that it COV2-AC still inhibits subsequent AC uptake (Figure 4F), suggesting that different mechanisms modulate macrophage inflammatory cytokine production and efferocytosis repression in response to COV2-AC.

2. What would be the effect of the COVID19 vaccine? Brazil reached over 55% or fully vaccinated population. Could the authors evaluate the expression of efferocytosis genes in PBMCs from healthy, COVID19 infected unvaccinated, COVID19 infected vaccinated? The same format of the experiment as Figure 4E.

We appreciate this assessment and agree with the reviewer that this is an interesting point. As we agreed with the editor’s and reviewer #1 suggestion to remove data on PBMC, this point can be better explored in a follow-up study.